# A RECIPE FOR SCALABLE ATTENTION-BASED ML POTENTIALS: UNLOCKING LONG-RANGE ACCURACY WITH ALL-TO-ALL NODE ATTENTION

## ABSTRACT

Machine-learning interatomic potentials (MLIPs) have advanced rapidly, with many top models relying on strong physics-based inductive bias, including rotational equivariance, high-order directional features, and energy conservation. However, as these MLIP models are being trained and evaluated on larger and larger systems, such as biomolecules and electrolytes, it is increasingly clear that solutions are needed for scalable and accurate approaches to long-range (LR) interactions in large systems. The most common approaches in literature to address long-range interactions rely on adding explicit physics-based inductive biases into the model. In this work, we propose a conceptually straightforward, data-driven, attention-based, and energy conserving MLIP, AllScAIP, that addresses long-range interactions and scales to O(100 million) training set sizes: a stack of local neighborhood self-attention followed by all-to-all node attention for global interactions across an entire atomistic system. Extensive ablations across model and dataset scales reveal a consistent picture: in low-data/small-model regimes, inductive biases help with improving some sample efficiency, and the all-to-all node attention increases LR accuracy. As data and parameters scale, the marginal benefit of these inductive biases diminishes (and can even reverse), while the all-to-all node attention remains the most durable ingredient for learning LR interactions. Our model achieves state-of-the-art on both energy/force accuracy and relevant physics-based evaluations on a representative molecular dataset (OMol25), while being competitive on materials (OMat24), and catalyst (OC20) datasets.

## 1 INTRODUCTION

Machine-learning interatomic potentials (MLIPs) learn surrogate potential-energy surfaces from *ab initio* data to enable molecular dynamics at near–DFT fidelity and practical cost (Unke et al., 2021). The field has progressed from descriptor-based neural potentials (Behler & Parrinello, 2007) to invariant message-passing networks (Schütt et al., 2017; Gasteiger et al., 2021; 2022), SE(3)-equivariant architectures with tensor features (Batzner et al., 2022; Passaro & Zitnick, 2023; Fu et al., 2025), and more recently attention-based models (Orbital-Materials, 2024; Qu & Krishnapriyan, 2024). Across these approaches, designs differ chiefly in their inductive biases: symmetry (translation, permutation, and rotation invariance or equivariance), locality (cutoffs and neighbor stencils), smoothness/regularization, architectural constraints, and explicit physical structure (energy-conserving gradients). These choices reflect two perspectives: one emphasizes richer inductive biases to encode geometry and physics up front, which can gain sample efficiency on small datasets. The other treats some inductive biases as a scaffold that can be reduced as data and parameters grow, letting the model discover features end-to-end while prioritizing training efficiency and stability.

As MLIPs have grown more capable, evaluation has shifted to larger and more complex systems. In these regimes, such as biomolecules and electrolytes systems, long-range (LR) interactions are important for describing subtle and important interactions. LR effects, such as electrostatics, induction/polarization, dispersion, electronic structure changes, couple atoms over many hops and across long length scales. However, the most scalable MLIPs for very large datasets are message-passing networks built on local radius-graph constructions with distance cutoffs. A common approach to address these limitations is to add explicit physics-based inductive biases (Unke et al., 2021): predict per-atom charges (or

multipoles) and evaluate Coulomb terms via Ewald/PME or FMM (Cheng, 2025; Kim et al., 2025); attach polarization or charge-equilibration solvers (Ko et al., 2021a); incorporate analytic or learned dispersion (e.g., many-body vdW) (Sauer et al., 2025); or introduce continuum/elastic corrections (Gong et al., 2025). These strategies have been extensively validated on a number of targeted, small-scale datasets. However, as the field shifts towards models that can deliver high accuracy across large, heterogeneous datasets spanning many distinct systems, driven in part by the release of large-scale datasets, developing approaches that scale effectively to this setting remains an open challenge.

We hypothesize that several inductive biases are *learnable under scale*—notably rotational symmetry, high-order directional features, and long-range interactions—whereas others are harder to learn and may need to be encoded by the architecture (locality via radius graph structure; energy conservation via gradient-based forces). Guided by this view, we develop a conceptually straightforward, scalable, attention-based MLIP with two stages of operations (illustrated in Fig 1): *neighborhood self-attention* (based on the EScAIP model in Qu & Krishnapriyan (2024)) operating on fixed local stencils to resolve local information, followed by an *all-to-all node self-attention* that mixes information globally, allowing signals to travel over the whole graph. Both stages use off-the-shelf multi-head self-attention operation CUDA kernels that have been highly engineered and optimized for popular AI/ML applications in computer vision and language. To test our hypothesis, we add two more ingredients to the recipe that provide features not directly captured by the architecture itself: *Legendre Angular Encoding (LAE)*, a compact, rotation-aware edge encoding that supplies high-order directional information to the neighborhood attention; and *Euclidean Rotary Position Encoding (RoPE)* (based on Frank et al. (2024)), an isotropic distance-only encoding that injects distance information into the node attention. These serve as inductive biases that can be explicitly built into the model architecture. We ablate these components across model and data scales to test whether such geometric inductive biases are indeed learnable when model capacity and data size grow.

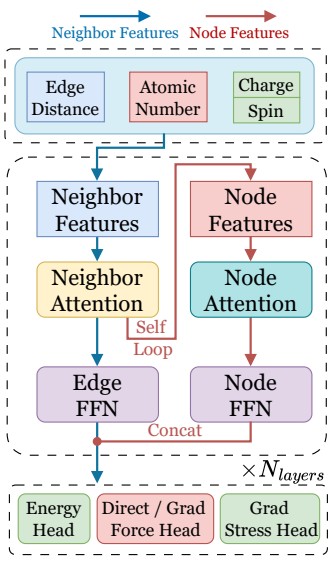

Figure 1: **AllScAIP model design.** The simple backbone design enables efficient scaling.

Across datasets and scales, the ablations support the "inductive biases learnable under scale" hypothesis. In the low-data/small-model regime, every ingredient is useful: adding LAE lowers force error by supplying directional/angle signals, the Euclidean RoPE improves energy with distance information, and the global node attention delivers the largest gains by enabling many-hop communication without deep stacks. As we increase both model capacity and dataset size, the picture shifts: the marginal benefit of the geometric encodings contracts toward zero and sometimes reverses sign, indicating that angular and radial features can be learned and absorbed end-to-end when scale is available. By contrast, the all-to-all node-attention stage remains the most durable source of long-range improvement.

In summary, our results support a *prior-light* recipe for scalable MLIPs. With sufficient data and parameters, several inductive biases—rotational equivariance, high-order directional patterns, and even long-range interactions—are largely learnable; lightweight geometric cues (LAE, Euclidean RoPE) help in low-data/small-model regimes but their marginal value fades—and can reverse—under scale. The full design—local neighborhood attention followed by all-to-all node-attention—uses off-the-shelf multi-head self-attention kernels. The resulting model: AllScAIP (**All**-to-all **S**calable **A**ttention **I**nteratomic **P**otential), attains state-of-the-art on both energy/force accuracy and relevant physics-based evaluations on representative molecules (OMol25) dataset, while being competitive on materials (OMat24) and catalysts (OC20) datasets. Taken together, the results suggest that a data-driven path for MLIPs may be competitive with other approaches: prioritize scalable components, keep priors lightweight, and letting scale handle the rest.

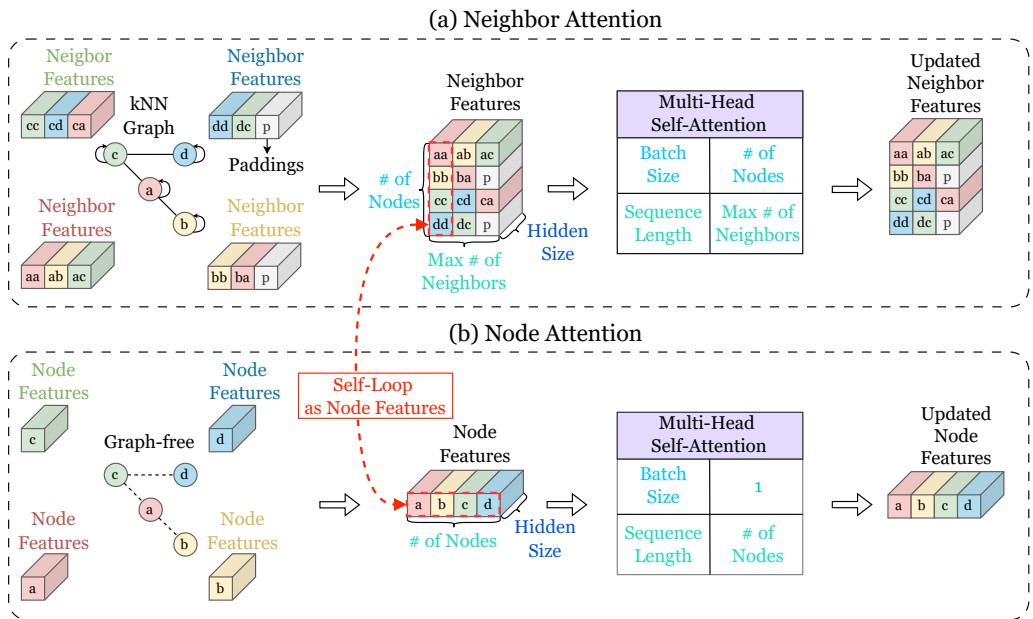

Figure 2: Illustration of the attention operations used in AllScAIP. (a) Neighborhood attention. (b) Node attention.

## 2 RELATED WORKS

**Machine Learning Interatomic Potentials.** There have been significant advances in neural network interatomic potentials (NNIPs), which are machine learning models that are trained to predict energies and per-atom forces from system descriptors such as atomic numbers and positions (Yuan et al., 2025). We group current approaches into two categories: (1) models that use equivariant node features (more inductive biases), including NeuqIP (Batzner et al., 2022), MACE (Batatia et al., 2022), SCN (Zitnick et al., 2022), eSCN (Passaro & Zitnick, 2023), Equiformer (Liao & Smidt, 2022; Liao et al., 2024), eSEN (Fu et al., 2025), UMA (Wood et al., 2025); (2) models that use scalar node features, where they only enforce basic symmetries, such as rotation and translation invariance: SchNet (Schütt et al., 2017), DimNet (Gasteiger et al., 2020), GemNet (Gasteiger et al., 2021; 2022), EScAIP (Qu & Krishnapriyan, 2024), OrbNet (Orbital-Materials, 2024).

**Long-range Interactions in MLIPs.** To enable the ability to model long range interaction in local GNN-based MLIPs, the main paradigm is to inject explicit long-range physics: models predict charges, multipoles, or surrogate charge densities and evaluate electrostatics with Ewald/PME/FMM, including PhysNet (Unke & Meuwly, 2019), 4G-HDNNP (Ko et al., 2021b), and DPLR (Zhang et al., 2022). Recent "latent" approaches avoid label requirements by learning a hidden per-atom variable and applying an Ewald-style long-range energy directly (LES) (Cheng, 2025). Dispersion has likewise been incorporated via machine-learned many-body vdW (Tu et al., 2023).

## 3 METHODS

### 3.1 ATTENTION OPERATIONS

We treat attention as an off-the-shelf operation, taking advantage of the highly optimized CUDA kernels (Lefaudeux et al., 2022). The only difference between the two stages is how we *pack tokens*: local (neighbor) attention operates on fixed neighbor list per node and scales as $\mathcal{O}(Nk)$, while all-to-all node attention mixes all nodes per graph with $\mathcal{O}(N^2)$ cost.

**Neighborhood Self-attention.** Following the EScAIP model (Qu & Krishnapriyan, 2024), each center atom gathers up to $k$ neighbors plus a self token, yielding a tensor of shape (#nodes) × $(k+1) \times d_{\text{model}}$ that we feed to standard multi-head self-attention (MHSA) (Fig 2 (a)). We run two

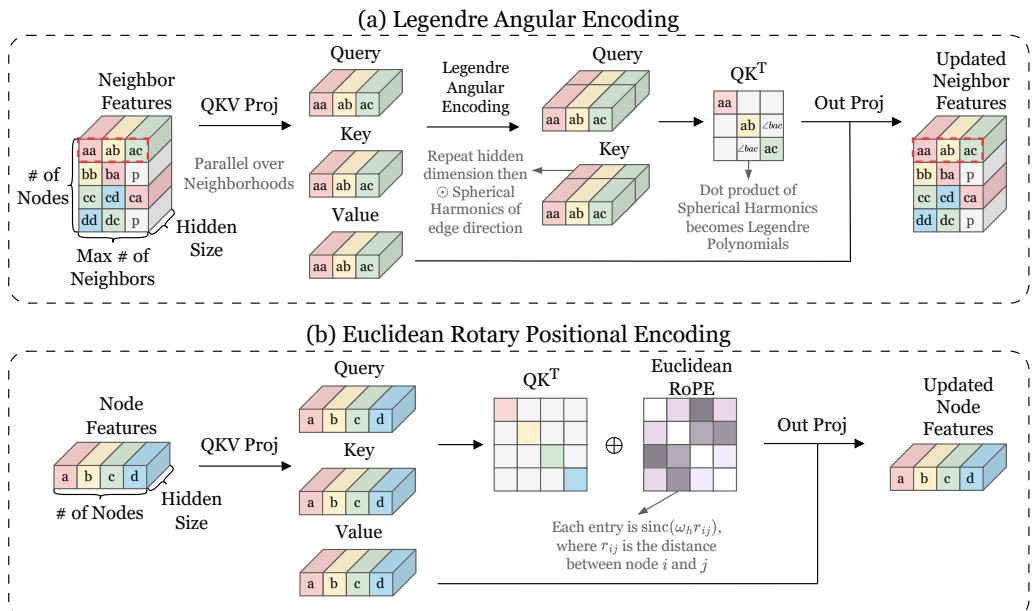

Figure 3: Illustration of the geometric encoding used in AllScAIP. (a) Legendre Angular Encoding (LAE) (b) Euclidean rotary position encoding.

directional passes over the same stencil: center→neighbors (out) and neighbors→center (in). A smooth distance-based envelope provides a padding/mask that softly fades far pairs.

**All-to-all Node Self-attention.** Local neighborhoods resolve fine geometry, but long-range interactions require global communication. We therefore apply MHSA over the node stream by packing the nodes to $1 \times (\#\text{nodes}) \times d_{\text{model}}$ and using the same operator (Fig 2 (b)). In practice, this stage complements the local passes: local attention handles fine, anisotropic interactions; node attention enables many-hop coupling in a single step.

## 3.2 GEOMETRIC ENCODINGS

In both neighborhood and node attention, each entry in the attention matrix corresponds to a geometric pair: an edge–edge interaction in the neighborhood case and a node–node interaction in the global case. We exploit this by injecting geometric encoding into the attention logits: *Legendre Angular Encoding* (LAE) for Spherical Harmonics-based high-order directional features in the neighborhood attention, and *Euclidean RoPE* (based on Frank et al. (2024)) for radial and distance-based features in the node attention (Figure 3). These encodings provide directional (angles) and radial (distances) signals that are not explicitly present in the vanilla Q/K/V streams; in principle, a sufficiently large model could infer them from coordinates and species. It's an important choice whether to include them as input (inductive biases) or let the model directly learn from data/scaling. See Appendix A for detailed description of both encodings.

## 3.3 ALLSCAIP MODEL

Figure 1 assembles the four components into a single block, stacked $N_{\text{layers}}$ times. In the input block, the node features are initialized from atom numbers (with optional charge/spin) and broadcast to form edge features together with distances RBF. Each block first applies neighborhood attention on the neighbor stream (with optionally added LAE encoding). This local stage is followed by RMSNorm, residual path, and an edge FFN. The block then takes self-loop neighbor features as node features, and performs all-to-all node attention on the node stream (with optional Euclidean RoPE), followed by RMSNorm, residual path, and a node FFN. The updated node features are concatenated with the neighbor features and carried to the next layer. After $N_{\text{layers}}$ blocks, the node stream feeds an energy head; forces are supervised either directly or via energy gradients (energy-conserving option), and a gradient-based stress head is included when required.

## 3.4 INDUCTIVE BIASES

We distinguish **hard** priors we encode by architecture, **soft** priors we can optionally supply as features, and **learnable** physical structure that we leave to scale. The goal is a prior-light model that keeps only what is hard to learn with high precision (locality, conservation, permutation/translation symmetries) while letting large models and datasets learn the rest. Here are the inductive biases we have considered:

| Inductive Biases | Status | How satisfied / where |
|---|---|---|
| Translation invariance | **Enforced** | Use only relative geometry (distances/directions). |
| Permutation equivariance | **Enforced** | Attention over sets with shared weights. |
| Extensivity (additivity) | **Enforced** | Sum aggregation for energy and PBC distance. |
| Locality prior | **Enforced** | kNN graph with smooth cutoff in neighborhood attention. |
| Energy conservation | **Enforced**[†] | Grad-based forces: $F = -\mathrm{d}E/\mathrm{d}x$; differentiable kNN graph. |
| Rotational equivariance | *Learnable* | Not hard-coded; can be learned under scale. |
| Long-range effects | *Learnable* | All-to-all node attention provides an infinite receptive field. |
| Radial/distance prior | *Optional (soft)* | **Euclidean RoPE**: distance-based encoding in node attention. |
| High-order directional features | *Optional (soft)* | **LAE**: compact spherical harmonic encoding modulates Q/K in neighborhood attention (angles/directions). |

[†]When using the conservative setting (our default): forces are obtained as energy gradients. A direct-force head is also supported for ablations.

In short, we enforce core symmetries (translation, permutation), locality, energy conservation, and extensivity; we optionally inject light geometric features; and we leave rotation and long-range interaction to be learned with scale. We validate these properties in § 5.2.

## 4 ABLATIONS

**Ablations on Model Components.** We begin by isolating the contribution of each architectural piece at a fixed capacity and data scale. We use the medium configuration (AllScAIP-md, about 85M parameters) trained on OMol25 4M split (Levine et al., 2025) for 80 epochs with direct force supervision. We toggle neighborhood self-attention (NeiAtt; always on), Legendre Angular Encoding (LAE), the all-to-all node attention (NodeAtt), and the Euclidean radial bias (ERoPE). We report Energy/Atom MAE (mEV) and Force MAE (meV/Å) on the validation set for total and four splits: Biomolecules, Electrolytes, Metal Complexes, and Neutral Organics (Table 1, top 4M section).

Removing LAE consistently worsens the force performance across all splits, while its effect on energies is smaller. This matches the role of LAE as a directional/angle feature: adding orientation information improves how the model fit to vector targets. In contrast, turning off ERoPE primarily harms energies and has a milder impact on forces, consistent with ERoPE's distance-only, isotropic bias acting as a radial prior on scalar predictions. Finally, ablating the all-to-all node-attention degrades both energy and force, with especially pronounced gaps in biomolecules, where systems are larger and the long-range effects are stronger. The full configuration achieves the best results across all splits, confirming that the modules we added improves sample efficiency in the low data regime.

**Ablations on Data and Model Size Scaling.** To quantify how inductive bias interacts with scale, we vary two axes: data (OMol 4M → OMol 102M) at fixed capacity, and model size (35M → 85M) at fixed data.

Under **data scaling** (4M → 102M) at fixed capacity, energy and force errors decrease across domains. In many cases, removing the fixed geometric encodings matches or slightly improves upon the full model at the same scale, suggesting that angular/radial signals can be learned directly from larger data. The all-to-all node attention remains consistently useful, indicating that a global mixing stage is a durable mechanism for long-range interactions.

Under **model-size scaling** (35M → 85M) at fixed data, absolute errors also drop with capacity. At 4M, geometric encodings still help, but their benefit diminishes as model size grows; at 102M, medium models show parity or slight disadvantages for fixed encodings, while retaining a measurable advantage

Table 1: **Component ablations under data and model size scaling**. We reoport Energy / Atom MAE (meV) and Force MAE (meV/Å), lower is better. Each size's **with encoding** (Nei Att + LAE + Node Att + ERoPE) configuration is the reference; other rows are colorized relative to that reference (Green = better, Red = worse). The Throughput column reports inference speed (ns/day on one H100 with 1000 atoms). We report results for the OMol25 4M split (80 epochs) and the full 102M (10 epochs).

| | | Ablations | | | | OMol 4M (80 Epochs) | | | | | | | | | |
| | | | | | | Biomol. | | Elytes. | | Metal Cplx. | | Neutral Org. | | Total | |
| Size | Thrpt | NeiAtt | LAE | NodeAtt | ERoPE | E↓ | F↓ | E↓ | F↓ | E↓ | F↓ | E↓ | F↓ | E↓ | F↓ |
|---|---|---|---|---|---|---|---|---|---|---|---|---|---|---|---|
| 35M | 2.279 | ✓ | ✓ | ✓ | ✓ | 0.38 | 5.57 | 1.27 | 8.93 | 2.85 | 33.52 | 1.89 | 13.52 | 1.46 | 9.35 |
| | 7.623 | ✓ | | ✓ | | 0.5 | 6.32 | 1.51 | 9.81 | 2.96 | 34.56 | 2.23 | 14.51 | 1.75 | 10.21 |
| 85M | 1.124 | ✓ | ✓ | ✓ | ✓ | 0.23 | 3.51 | 1.05 | 6.38 | 2.46 | 27.83 | 1.21 | 8.93 | 1.16 | 6.88 |
| | 3.333 | ✓ | | ✓ | ✓ | 0.29 | 4.27 | 1.17 | 7.22 | 2.67 | 29.79 | 1.53 | 10.53 | 1.22 | 7.6 |
| | 1.392 | ✓ | ✓ | ✓ | | 0.29 | 3.78 | 1.16 | 6.67 | 2.58 | 28.26 | 1.36 | 9.03 | 1.3 | 7.1 |
| | 4.014 | ✓ | | ✓ | | 0.32 | 4.24 | 1.17 | 7.16 | 2.64 | 29.19 | 1.57 | 10.12 | 1.25 | 7.55 |
| | 1.281 | ✓ | ✓ | | | 0.52 | 4.58 | 1.39 | 7.42 | 2.81 | 29.86 | 1.43 | 10.01 | 1.56 | 7.93 |
| | 4.327 | ✓ | | | | 0.55 | 4.96 | 1.45 | 7.92 | 2.85 | 30.78 | 1.57 | 10.77 | 1.72 | 8.48 |
| Size | Thrpt | NeiAtt | LAE | NodeAtt | ERoPE | OMol 102M (10 Epochs) | | | | | | | | | |
| 35M | 2.279 | ✓ | ✓ | ✓ | ✓ | 0.21 | 3.85 | 0.75 | 6.01 | 2.09 | 26.02 | 1.01 | 8.54 | 0.85 | 6.61 |
| | 7.623 | ✓ | | ✓ | | 0.29 | 4.47 | 0.81 | 6.73 | 2.12 | 26.86 | 1.08 | 9.17 | 0.98 | 7.22 |
| 85M | 1.124 | ✓ | ✓ | ✓ | ✓ | 0.15 | 2.81 | 0.53 | 4.59 | 1.83 | 21.93 | 0.73 | 6.03 | 0.67 | 5.1 |
| | 3.333 | ✓ | | ✓ | ✓ | 0.2 | 3.16 | 0.55 | 4.99 | 1.83 | 22.48 | 0.79 | 6.47 | 0.73 | 5.51 |
| | 1.392 | ✓ | ✓ | ✓ | | 0.2 | 2.93 | 0.58 | 4.76 | 1.8 | 21.59 | 0.76 | 5.94 | 0.72 | 5.13 |
| | 4.014 | ✓ | | ✓ | | 0.15 | 2.91 | 0.52 | 4.7 | 1.83 | 22.31 | 0.72 | 6.29 | 0.64 | 5.23 |
| | 4.327 | ✓ | | | | 0.4 | 3.83 | 0.79 | 5.57 | 2.01 | 23.91 | 0.85 | 7.05 | 0.96 | 6.15 |

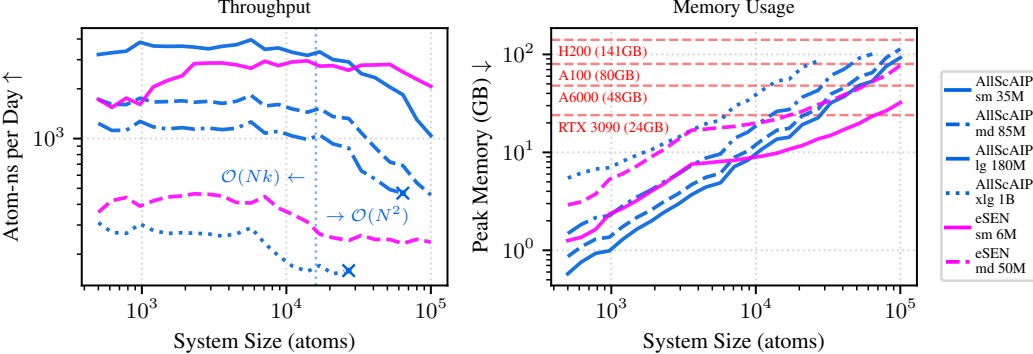

Figure 4: **Throughput and Memory vs. System size.** Atom-time (number of atoms × ns/day, higher is better) is measured on a single H200 141G with graph generation off. Left line show four model sizes (35M/85M/180M/1B) of AllScAIP, and eSEN baselines. The dotted vertical lines indicate approximately when the $\mathcal{O}(N^2)$ of the node attention dominates over the $\mathcal{O}(Nk)$ of the neighborhood attention, where $k$ is the max number of neighbors. Right show the vram usage, and the vram size for common GPUs.

from the all-to-all attention. In short: as data and parameters scale, the marginal value of fixed geometric encodings decreases, whereas the architectural affordance of global mixing continues to pay off.

## 5 RESULTS

### 5.1 INFERENCE EFFICIENCY AND SYSTEM–SIZE SCALING

We test the efficiency and memory scaling of the AllScAIP model. Figure 4 plots atom–ns/day versus system size on a single H200 141G (graph generation off) that exposes a slope change. In the small $N$ regime, speed is dominated by neighborhood attention ($\mathcal{O}(Nk)$), atom–ns/day is nearly flat or linear in $N$; once the all-to-all node attention dominates ($\mathcal{O}(N^2)$), it falls roughly as $1/N$ (slope $-1$ in log-log). The dashed vertical lines mark this empirical transition. Overall, the curves show predictable scaling with a clear regime switch, and our models are still efficient compared with baselines. Our method targets around $10^3 - 10^5$ atoms, where much interesting science lives (biomolecules, electrolytes, soft matter, mesoscale crystallites) and GPUs remain efficient. In addition, we also provide the raw

Table 2: **Symmetry and Conservation Checks.** Extensivity errors for periodic (PBC) and non-periodic (non-PBC) systems, (lower is better), rotational equivariance via force cosine similarity under random rotations (higher is better), and NVE molecular dynamic simulation energy drift on MD22 large molecules (lower is better).

| Model | Training Set | PBC Energy $\Delta$ [meV] $\downarrow$ | non-PBC Energy $\Delta$ [meV] $\downarrow$ | Rand. Rotation Cos. Sim. $\uparrow$ | NVE MD Energy Drift [meV / atom / ps] $\downarrow$ |
|---|---|---|---|---|---|
| UMA-M-1p1 | UMA-459M | $2.41 \times 10^{-2}$ | $3.83 \times 10^{-3}$ | 0.9999 | $1.5 \times 10^{-3}$ |
| AllScAIP-md | NA (Random Init) | $3.62 \times 10^{-2}$ | $2.49 \times 10^{-3}$ | 0.6827 | - |
| AllScAIP-md | OMol-102M | $4.18 \times 10^{-2}$ | $4.73 \times 10^{-3}$ | 0.9999 | $4.3 \times 10^{-3}$ |

throughput vs. system size at Figure 11, and per-component (Nei Att/FFN, Node Att/FFN) breakdowns and wall-clock measurements at Figure 10.

## 5.2 SYMMETRY & CONSERVATION CHECKS

We verify that AllScAIP satisfies key inductive-bias properties and compare to a strong baseline trained on various chemical systems—UMA (Wood et al., 2025) (Table 2).

**Extensivity.** We test extensivity in both periodic and non-periodic settings:

(i) **PBC supercell doubling:** starting from a periodic structure, we build a $2\times$ supercell by translating the cell by one lattice vector and recomputing the energy; extensivity requires $E_{2\times} \approx 2E_{1\times}$. We report the $E_\Delta = |E_{2\times} - 2E_{1\times}|$.

(ii) **Vacuum duplication:** for a non-periodic system, we duplicate the configuration and translate the copy by a large offset $R$, then evaluate $E(R)$; as $R \to \infty$ the fragments are non-interacting and $E(R) \to 2E_{\text{single}}$. In practice, we use $R = 1000$Å and report the deviation $E_\Delta = |E(R) - 2E_{\text{single}}|$.

Because the architecture uses only relative geometric features in attention and no absolute positional codes, it shows near-perfect additivity under both tests, confirming that the learned long-range coupling does not introduce unphysical behavior.

**Rotational Equivariance.** We sample 1000 OMol test structures, draw a random rotation $R \in \mathrm{SO}(3)$, and evaluate forces twice: $\mathbf{F}^{(1)}$ on the original coordinates and $\mathbf{F}^{(2)}$ on the rotated atom position $R\mathbf{X}$. We then rotate the first prediction, $R\mathbf{F}^{(1)}$, and compute the per-atom cosine similarity with $\mathbf{F}^{(2)}$, averaging over atoms and structures. AllScAIPtrained on OMol-102M achieves a cosine similarity of 0.9999, on par with UMA, whereas a randomly initialized model yields a much lower value. This indicates that rotational equivariance is *learned*.

**Energy Conservation.** Following Fu et al. (2025), we run NVE molecular dynamics simulation on the seven large MD22 molecules (Chmiela et al., 2023) for 100 ps and report the energy drift (eV/atom/ps). AllScAIP shows small drift, comparable in scale to the UMA model.

## 5.3 OPEN MOLECULES (OMOL25)

**Settings.** We train AllScAIP on the OMol25 (Levine et al., 2025) dataset under the following settings:

a. **4M split:** (i) direct force training for 80 epochs (**–d.**); (ii) a conservative fine-tune where we train direct force for 50 epochs, then swap to a gradient-based energy / force head and continue for 30 epochs (**–ft-cons.**).

b. **Full 102M:** (i) direct force for 10 epochs (**–d.**); (ii) direct force for 10 epochs + 2-epoch conservative fine-tune (**–ft-cons.**); (iii) fully conservative training from scratch (**–cons.**).

All models are trained on V100 32G with full fp32.

**Energy and Force Accuracy.** Across OMol25, our models sit on the accuracy–speed Pareto (Figure 5) and deliver state-of-the-art energy error with competitive force error (Table 3). On the 4M split, the conservative fine-tuned medium model attains the best overall energy while remaining close in force to the strongest baseline; the direct-force variant trades a small energy increase for better forces. On the full 102M split, the medium direct force model achieves the best overall energy accuracy. The same trends hold across all four splits, with especially strong gains on **biomolecules**-the largest systems in OMol-where long-range interactions is most critical.

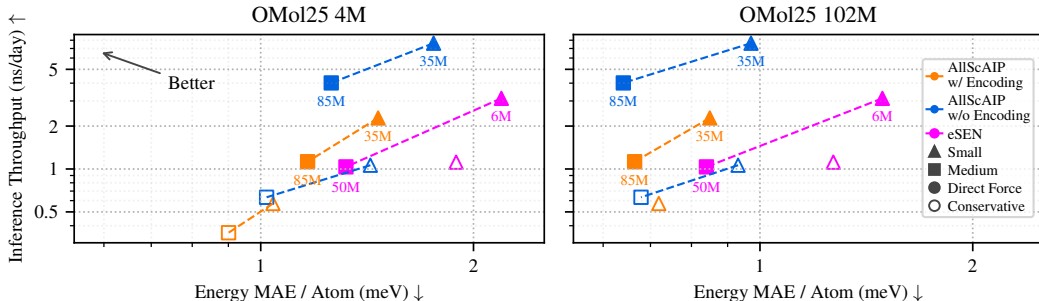

Figure 5: **OMol25 energy error vs. throughput.** Energy / Atom MAE (meV, ↓) vs. throughput (ns/day, ↑) for 35M/85M models with/without encodings; compared with eSEN (Fu et al., 2025) baselines. Conservative models are labeled with a hollow marker. Left: Models trained on OMol25 4M for 80 epochs. Right: OMol25 102M (12 epochs). Our models trace the Pareto front at 4M; at 102M the gap between with/without encodings shrinks or flips, indicating these inductive bias may be unnecessary at scale. The larger speed gap between the direct force and conservative AllScAIP models, compared to eSEN, occurs because the differentiable kNN graph construction used in AllScAIP is a newly introduced operation that has not yet been fully optimized for differentiation.

Table 3: **OMol25 validation results.** Energy / Atom MAE (meV) and Force MAE (meV/Å) across Biomolecules, Electrolytes, Metal Complexes, and Neutral Organics. AllScAIP variants achieve the lowest overall energy error on both splits, with competitive force errors.

| Dataset | Model | Biomolecules Energy ↓ | Biomolecules Force ↓ | Electrolytes Energy ↓ | Electrolytes Force ↓ | Metal Complexes Energy ↓ | Metal Complexes Force ↓ | Neutral Organics Energy ↓ | Neutral Organics Force ↓ | Total Energy ↓ | Total Force ↓ |
|---|---|---|---|---|---|---|---|---|---|---|---|
| All | eSEN-sm-d. | 0.67 | 6.30 | 1.24 | 9.41 | 2.53 | 33.08 | 1.23 | 13.84 | 1.49 | 9.92 |
| | eSEN-sm-cons. | 0.59 | 4.61 | 1.01 | 8.08 | 2.30 | 28.86 | 0.84 | 11.11 | 1.27 | 8.25 |
| | eSEN-md-d. | 0.34 | **2.61** | 0.69 | **4.40** | **1.73** | **19.99** | **0.59** | **5.63** | 0.84 | **4.76** |
| | GemNet-OC | **0.15** | 3.88 | 0.56 | 5.98 | 1.83 | 25.12 | 0.86 | 10.38 | 0.66 | 6.52 |
| | AllScAIP-sm-ft-cons. | 0.22 | 3.84 | 0.75 | 6.01 | 2.09 | 26.02 | 1.00 | 8.55 | 0.85 | 6.61 |
| | AllScAIP-md-d. | **0.15** | 2.91 | **0.52** | 4.70 | 1.83 | 22.31 | 0.72 | 6.29 | **0.64** | 5.24 |
| | AllScAIP-md-cons. | 0.22 | 3.23 | 0.53 | 5.29 | 1.86 | 22.40 | 0.59 | 7.55 | 0.67 | 5.78 |
| 4M | eSEN-sm-d. | 0.88 | 8.12 | 1.93 | 12.64 | 3.37 | 40.44 | 2.16 | 20.17 | 2.19 | 13.01 |
| | eSEN-sm-cons. | 0.86 | 6.17 | 1.61 | 11.16 | 2.72 | 35.33 | 1.50 | 16.92 | 1.89 | 11.10 |
| | eSEN-md-d. | 0.47 | 3.38 | 1.18 | **6.51** | 2.53 | **27.31** | 1.21 | **9.26** | 1.32 | **6.78** |
| | GemNet-OC-r6 | 0.40 | 5.84 | 1.39 | 9.37 | 2.74 | 33.60 | 1.88 | 16.55 | 1.41 | 9.83 |
| | GemNet-OC | 0.25 | 5.20 | 1.04 | 8.42 | 2.66 | 32.76 | 1.64 | 15.59 | 1.13 | 8.98 |
| | AllScAIP-sm-ft-cons. | 0.27 | 4.06 | 0.91 | 7.81 | 2.26 | 32.10 | 1.11 | 12.73 | 1.04 | 8.19 |
| | AllScAIP-md-ft-cons. | **0.20** | **3.01** | 0.75 | 6.79 | 2.06 | 29.41 | 0.78 | 9.45 | 0.90 | 7.67 |

Table 4: **Evaluation results across OMol25 test evaluations.** Results are reported in energy MAE error [meV] (lower is better). Models are sorted by the average ranking of individual categories, with a lower average ranking indicating better overall performance.

| Model | Energy Cons. | Training Set | Avg. Rank | Ligand pocket Ixn Energy ↓ | Ligand strain Strain Energy ↓ | Conformers ΔEnergy ↓ | Protonation ΔEnergy ↓ | Dist. scaling ΔEnergy ↓ | IE/EA ΔEnergy ↓ | Spin gap ΔEnergy ↓ |
|---|---|---|---|---|---|---|---|---|---|---|
| AllScAIP-md-d. | | OMol-102M | 3.14 | 29.496 | 4.149 | 4.985 | 16.104 | 10.082 | 143.035 | 350.825 |
| AllScAIP-md-cons. | ✓ | OMol-102M | 3.71 | 50.532 | 4.103 | 3.874 | 17.092 | 36.145 | 148.976 | 317.400 |
| eSEN-md-d. | | OMol-102M | 3.86 | 64.246 | 3.113 | 3.566 | 14.988 | 109.767 | 145.776 | 303.883 |
| UMA-M-1p1 | ✓ | UMA-459M | 4.43 | 76.778 | 2.962 | 2.532 | 18.457 | 138.094 | 136.386 | 335.058 |
| AllScAIP-sm-ft-cons. | ✓ | OMol-102M | 5.71 | 58.450 | 4.770 | 4.106 | 27.515 | 15.473 | 168.858 | 341.708 |
| AllScAIP-md-ft-cons. | ✓ | OMol-4M | 5.86 | 46.727 | 4.839 | 5.172 | 19.399 | 16.473 | 169.104 | 367.133 |
| AllScAIP-sm-ft-cons. | ✓ | OMol-4M | 7.86 | 78.260 | 4.559 | 5.343 | 24.614 | 23.492 | 198.899 | 401.358 |
| GemNet-OC-r12 | | OMol-102M | 8.86 | 19.373 | 8.381 | 12.049 | 31.645 | 77.074 | 177.457 | 373.831 |
| eSEN-sm-cons. | ✓ | OMol-102M | 9.57 | 147.261 | 4.656 | 4.487 | 21.770 | 196.964 | 222.827 | 391.469 |
| GemNet-OC-r6 | | OMol-102M | 10.00 | 47.022 | 9.554 | 11.479 | 36.925 | 80.683 | 199.721 | 370.975 |
| eSEN-md-d. | | OMol-4M | 10.14 | 98.337 | 4.508 | 5.369 | 24.975 | 111.063 | 240.212 | 430.753 |
| UMA-S-1p1 | ✓ | UMA-459M | 10.14 | 127.723 | 4.856 | 5.170 | 27.969 | 194.688 | 206.872 | 369.181 |
| GemNet-OC-r12 | | OMol-4M | 12.00 | 36.221 | 12.489 | 19.797 | 50.773 | 99.995 | 237.682 | 490.463 |
| mace-omol-L-0 | ✓ | OMol-102M | 13.29 | 297.404 | 7.954 | 6.782 | 25.107 | 244.834 | 338.795 | 438.891 |
| eSEN-sm-cons. | ✓ | OMol-4M | 13.43 | 243.883 | 5.850 | 6.811 | 35.754 | 232.839 | 293.120 | 478.242 |
| GemNet-OC-r6 | | OMol-4M | 14.00 | 96.043 | 13.270 | 23.355 | 52.160 | 83.296 | 329.993 | 559.080 |

**OMol25 Test Evaluations.** We benchmark on the full OMol25 evaluation suite (Table 4), sorting models by their average ranking across seven categories. AllScAIP-md-d. trained on the full 102M split tops by average rank. In particular, our model excels on the distance scaling (LR) test (about 90% reduction compared with the second best model): when molecules are uniformly compressed/stretched, energy error for AllScAIP remains low and flat, while eSEN and UMA degrade markedly under large stretches (Fig. 6). These results indicate that the proposed architecture transfers beyond the training distribution, providing robust long-range behavior while maintaining competitive accuracy on the

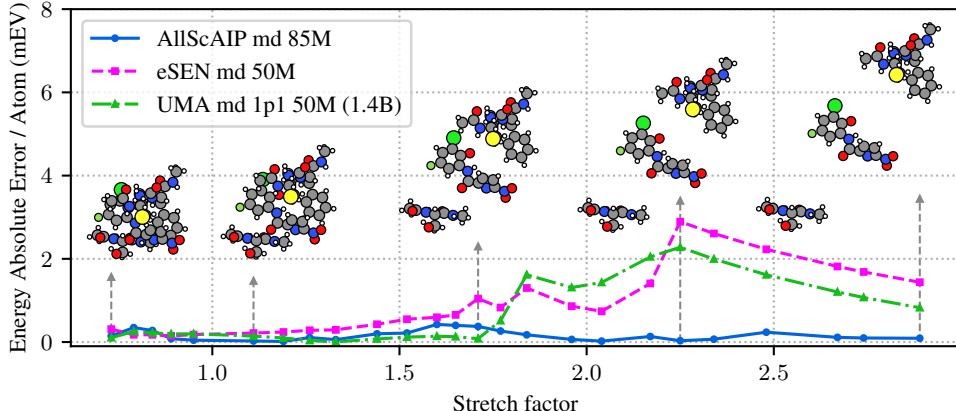

Figure 6: **OMol25 distance-scaling evaluation.** A group of molecules are uniformly compressed and stretched by a scalar "stretch factor" to probe the long-range ability (x-axis; $< 1$: compressed, $> 1$: stretched) and report energy absolute error per atom (meV; $\downarrow$). AllScAIP-md-cons. stays flat and low across the full range, while eSEN and UMA md-1p1 degrade sharply under stretching, indicating poor long-range capacities. Insets show example geometries at selected factors labeled with dotted lines.

remaining chemical benchmarks (ligand interaction/strain, conformers, protonation, IE/EA, and spin gaps). Details of the evaluations can be found in the OMol25 paper (Levine et al., 2025).

**MD Simulation and Comparison to Experiment.** To test whether energy and force accuracy transfers to macroscopic observables, we ran out-of-the-box NPT simulations with the OMol25-trained `AllScAIP-md` (no MD-specific fine-tuning). Stresses come from zero-shot gradient-based virial (not trained during training). We used a 1 fs timestep and ran NPT MD simulation for 300 ps, then computed densities and enthalpy of vaporization from the final 150 ps (after equilibration).

**a. Molecular liquids (39 liquids).** Following Kim et al. (2025), we ran NPT MD on 39 molecular liquid systems at 298K, and NVT MD on the corresponding isolated molecules to extract the potential energies for the enthalpy of vaporization (Figure 8). AllScAIP achieves better MAE and $R^2$ compared with MACELES. The eSEN points show a systematic over-prediction (compression), whereas AllScAIP removes this bias.

**b. Water vs temperature.** Following Kim et al. (2025), we simulated 64 water molecules across temperatures (Figure 7). AllScAIP tracks the experimental temperature dependence and values, while eSEN and MACE/MACELES over-predict densities across the range.

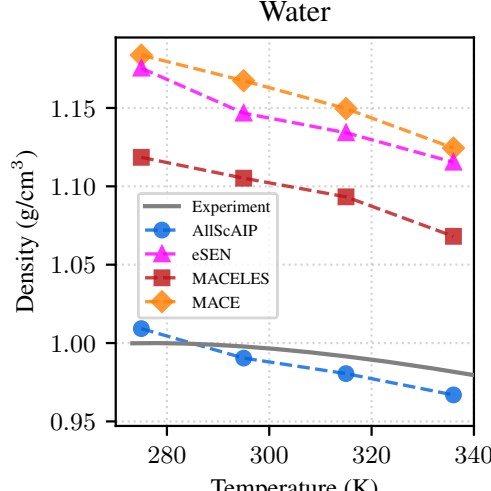

Figure 7: **MD Simulations vs. Experimental Density.** Density-temperature curve for a 64 water molecules system, from 300 ps NPT MD simulations. AllScAIP tracks the experimental trend and values, while eSEN, MACELES, and MACE over-predict density across temperatures.

Overall, the MD results indicate that the data-driven all-to-all node attention backbone yields realistic bulk behavior without explicit long-range terms, consistent with our long-range ablations and distance-scaling tests. Additional results on MD simulations with the MD22 (Chmiela et al., 2023) datasets can be found in the Appendix E

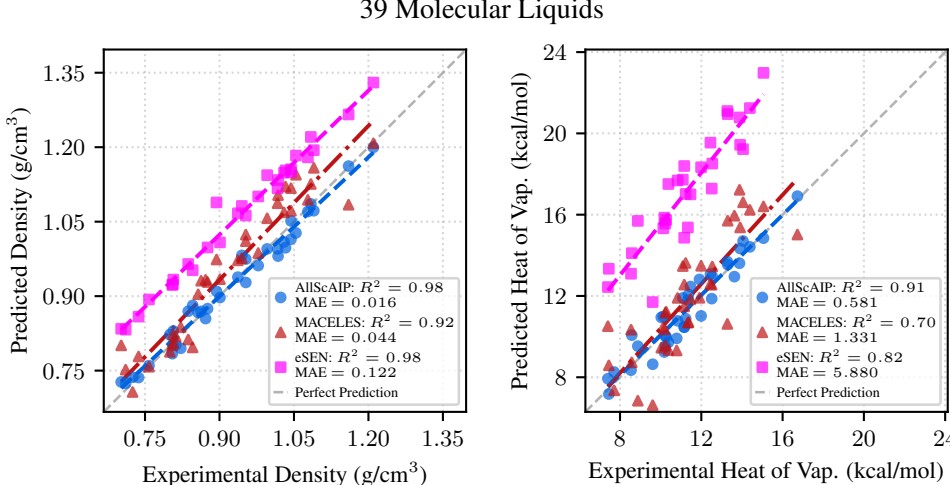

Figure 8: **MD Simulations vs. Experimental Density and Enthalpy of Vaporization.** NPT MD on 39 different molecular liquids and NVT MD on isolated molecules for 300 ps. Left: predicted vs. experimental densities; Right: predicted vs. experimental enthalpy of vaporization. AllScAIP attains lower MAE and higher $R^2$ than MACELES and eSEN on both properties.

## 5.4 OPEN MATERIALS (OMAT24) AND OPEN CATALYST (OC20)

AllScAIP achieves competitive performance on OMat24 (Barroso-Luque et al., 2024) and OC20 (Chanussot et al., 2021). See Appendix C for details.

## 6 DISCUSSIONS

**Scaling and the role of inductive bias.** Our ablations suggest a simple guideline: *scale first, bias second*. Directional/radial encodings (LAE, Euclidean RoPE) improve data efficiency in low-data or small-model regimes, but their marginal value shrinks—and can even reverse—as both data and parameters grow. In contrast, the architectural capacity for long-range coupling provided by the all-to-all node attention remains beneficial at every scale we tested. Practically, this argues for **prioritizing scalable, expressive components**—because data and compute will only increase; such components benefit monotonically with scale, whereas fixed inductive features may cap flexibility.

**Data-driven vs. high-order equivariant paths for MLIPs.** A parallel line of work encodes direction with SE(3)–equivariant spherical-harmonic (SH) irreps. This route provides a rich prior directional basis, but in practice it is *band-limited*: after truncating at degree $L$, very sharp or dataset-specific angular features (e.g. narrow hydrogen-bond cones, $\pi$-stacking orientations, etc.) require larger $L$. The representation is powerful yet rigid—capacity is tied to the chosen SH order and coupling rules. Our path is different: we keep the backbone scalar and data-driven. The model learns angular structure through attention. As scale grows, it can allocate capacity where needed without being constrained by an SH decomposition, preserving hardware efficiency and flexibility.

**Limitations and opportunities.** The efficiency study shows a clear slope change: throughput is flat/linear while local $\mathcal{O}(Nk)$ dominates and transitions to $\sim 1/N$ when the all-to-all $\mathcal{O}(N^2)$ node attention takes over. We view $\mathcal{O}(N^2)$ as a manageable trade-off for long-range accuracy, especially given the engineering playbook that enabled very long context in LLMs: tiling/chunked matmuls (Flash-style kernels), activation checkpointing, KV memory compression, and paging—most of which transfer directly to MHSA over atoms. Beyond systems work, several modeling routes can further delay the $\mathcal{O}(N^2)$ regime: hierarchical/coarse-grained node pools with cross-pool attention; linear-time attention; and mixtures-of-experts that route only a fraction of nodes to global mixing.

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

# A  ADDITIONAL DETAILS ON MODEL ARCHITECTURE

## A.1  GEOMETRIC ENCODINGS

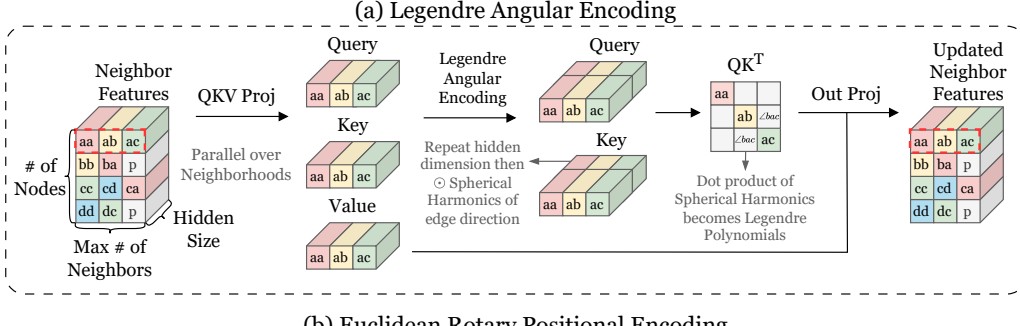

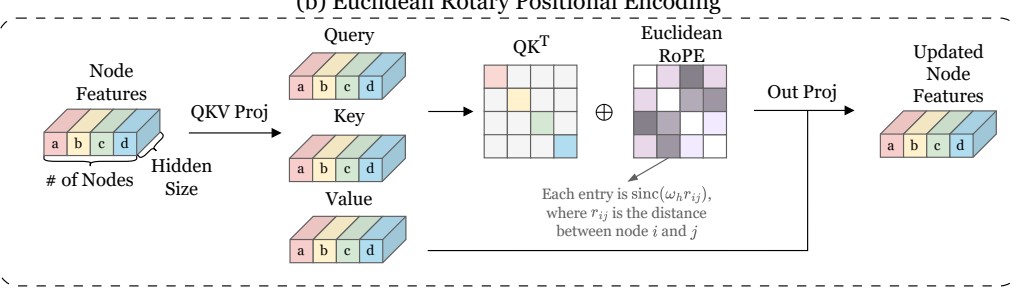

Figure 9: Illustration of the geometric encoding used in AllScAIP. (a) Legendre Angular Encoding (LAE) (b) Euclidean rotary position encoding.

**Legendre Angular Encoding (LAE).**  This encoding is used to inject angular and high-order directional features to the neighborhood attention logits. Each directed edge $i \to j$ is assigned a compact angular code built from real spherical harmonics evaluated on the unit direction $\hat{\mathbf{r}}_{ij}$. For degrees $\ell = 0,...,L$ we compute $\mathbf{Y}_\ell(\hat{\mathbf{r}}_{ij}) \in \mathbb{R}^{2\ell+1}$, repeat each degree block $r_\ell$ times, and concatenate across $\ell$ to obtain $\gamma_{ij} \in \mathbb{R}^{d_{h_r}}$ with $d_{h_r} = \sum_\ell r_\ell(2\ell+1)$. We choose $\{r_\ell\}$ such that the sum matches the per-head dimension: $\sum_{\ell=0}^{L} r_\ell = d_h$. For head $h$, the queries/keys are repeated $2\ell+1$ times accordingly and modulated elementwise by $\gamma_{ij}$,

$$\tilde{q}_{i,h} = \bar{q}_{i,h} \odot \gamma_{ij}, \quad \tilde{k}_{j,h} = \bar{k}_{j,h} \odot \gamma_{ij},$$

and the usual scaled dot-product is applied, $a_{ij,h} = (\tilde{q}_{i,h}^\top \tilde{k}_{j,h})/\sqrt{d_h}$, followed by standard attention. By the spherical-harmonic addition theorem, inner products within each degree correspond (up to a constant) to $P_\ell(\cos\theta)$, so LAE supplies multi-order angular structure with linear cost. LAE is rotation-aware, parameter-light, and integrates with off-the-shelf attention kernels (Figure 3 (a)).

**Euclidean Rotary Position Encoding (ERoPE).**  To inject distance information in the node attention attention logits, we add an isotropic radial bias. Following Frank et al. (2024), for each pair $(i,j)$ with separation $r_{ij}$, we build a purely radial encoding by averaging plane-wave phases over all orientations on the unit sphere, which yields the isotropic kernel $K_\omega(r) = \mathrm{sinc}(\omega r) = \sin(\omega r)/(\omega r)$. We choose a bank of $M$ log-spaced frequencies $\{\omega_k\}_{k=1}^M \subset [\omega_{\min}, \omega_{\max}]$ (shared across heads), evaluate $\mathbf{s}_{ij} = [\mathrm{sinc}(\omega_1 r_{ij}),...,\mathrm{sinc}(\omega_M r_{ij})]^\top$. Each head $h$ mixes them with a learned weight vector $\mathbf{w}_h \in \mathbb{R}^M$ to produce an additive logit bias $b_{ij,h} = \mathbf{w}_h^\top \mathbf{s}_{ij}$, and the final logit is

$$a_{ij,h} = \frac{q_{i,h}^\top k_{j,h}}{\sqrt{d_h}} + b_{ij,h}.$$

Because $b_{ij,h}$ depends only on $r_{ij}$, the bias is translation- and rotation-invariant by construction. The small frequency bank adds only $\mathcal{O}(M)$ work per pair, can be precomputed per batch, and provides a smooth, spectrally rich radial prior without altering the softmax mechanism (Figure 9 (b)).

# B   ADDITIONAL OMOL25 RESULTS

## B.1   EFFICIENCY

We report the runtime breakdown by component vs. system size in Figure 10. This profile explains the slope change observed in throughput vs. size curve. For modest N, compute scales like $\mathcal{O}(Nk)$ and is driven by neighborhood attention. Beyond a crossover (earlier for 180M, later for 35M), the all-to-all node attention $\mathcal{O}(N^2)$ component dominates. We note that the profiler adds time and memory overheads and we need the disable `torch.compile`, thus this could differ from the actual performance.

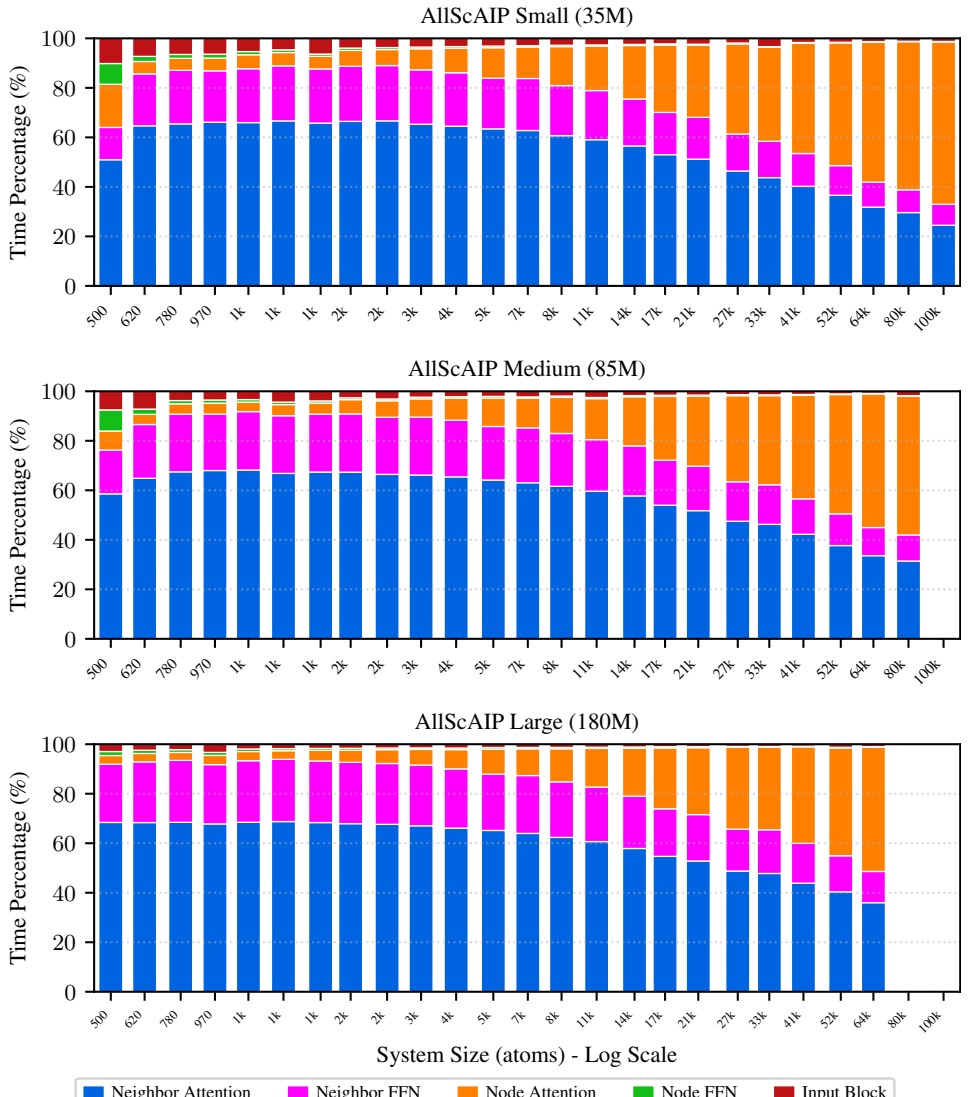

Figure 10: **Runtime breakdown by component vs. system size.** Stacked bars show the percentage of wall-clock inference time spent in each module of AllScAIP on a single H200 (graph construction off), for three model sizes (top: 35M, middle: 85M, bottom: 180M). Colors: Neighbor attention (blue, $\mathcal{O}(Nk)$), Neighbor FFN (magenta), Node attention (orange, $\mathcal{O}(N^2)$), Node FFN (green), and Input/embeddings (red). At small systems the runtime is dominated by the local block (neighbor attention). As the number of atoms grows into the tens of thousands, the cost of the global node attention increases and eventually becomes the majority of runtime; the crossover happens earlier for larger models.

We also provide a raw throughput vs. system size plot for additional comparison (Fig. 11):

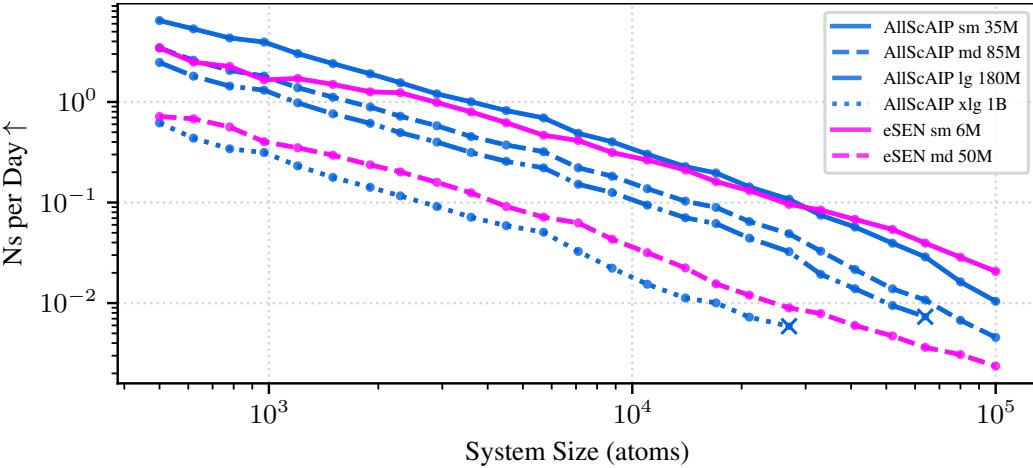

Figure 11: **Throughput vs. System size.** Throughput (ns/day, higher is better) is measured on a single H200 141G with graph generation off. Lines show four model sizes (35M/85M/180M/1B) of AllScAIP and eSEN baselines.

## B.2 LONG-RANGE VALIDATIONS ON OMOL25

To further investigate the long-range error for different models, we bin the OMol25 validation set by system diameter and total charge. We observe that AllScAIP's energy and force MAE remains lower, whereas eSEN's error grows significantly for large-diameter and highly charged systems (Figure 12).

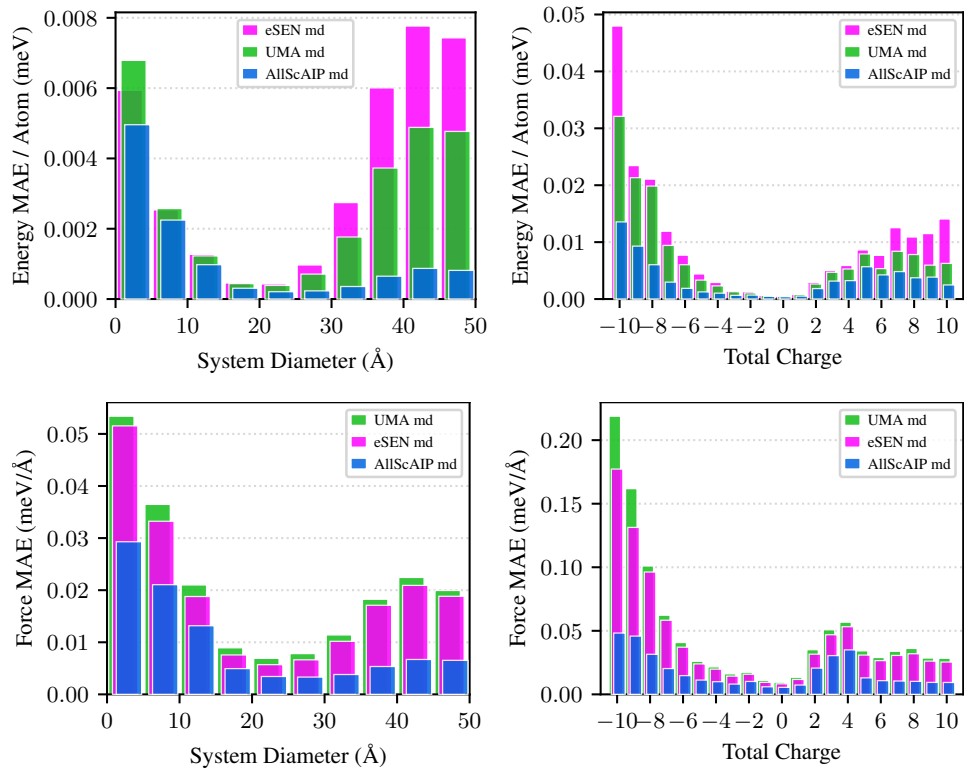

Figure 12: **OMol25 Prediction Error vs. System Diameter (Å) and Total Charge.** Energy / Atom MAE (meV, ↓, top) and Force MAE (meV/Å, ↓, bottom) binned by (left) system diameter (Å) and (right) total charge on all of OMol25 validation set. Bars compare AllScAIP-md, UMA-md, and eSEN-md. AllScAIP remains lower and more stable across bins, while eSEN and UMA error grows for large-diameter systems and for high total charge.

### B.3 VISUALIZATION OF ALL-TO-ALL NODE ATTENTION

To make the long-range behavior of the all-to-all node attention concrete, we visualize one representative large biomolecule (Figure 12, top-left) and extract attention weights from the final node-attention layer.

**Settings.** For the chosen structure we compute the pairwise distances $r_{ij}$ and collect the post-softmax attention matrices $A^{(h)} \in \mathbb{R}^{N \times N}$ for each head $h$. Distances are binned with $\Delta r = 0.5$Åover $[0,25]$Å.

**Mean attention vs. distance.** For each head we average the attention in each distance bin (row-wise mean, then averaged across rows) and plot the resulting curve. Figure 12 (top-right) shows all heads (thin) and the head average (thick). A vertical line indicates the local neighbor cutoff used to build the radius graph (6Å). We consistently observe that some heads place above-baseline (1/N) mass at large separations, while others remain local, yielding a flat head-average close to the uniform level.

**Distance and attention heatmaps.** The pairwise distance matrix confirms the contact structure of the system, while the head-averaged attention matrix reveals vertical bands ("hubs"): atoms that receive attention from many others, including at long distances. We fould some hubs coincide with chemically distinctive groups (e.g., charged or polar sites).

**Interpretation.** Taken together, the per-head distance profiles and the heatmaps indicate head specialization and long-range mixing in the node attention layer: a subset of heads routes information across tens of Å, complementing the local neighborhood attention.

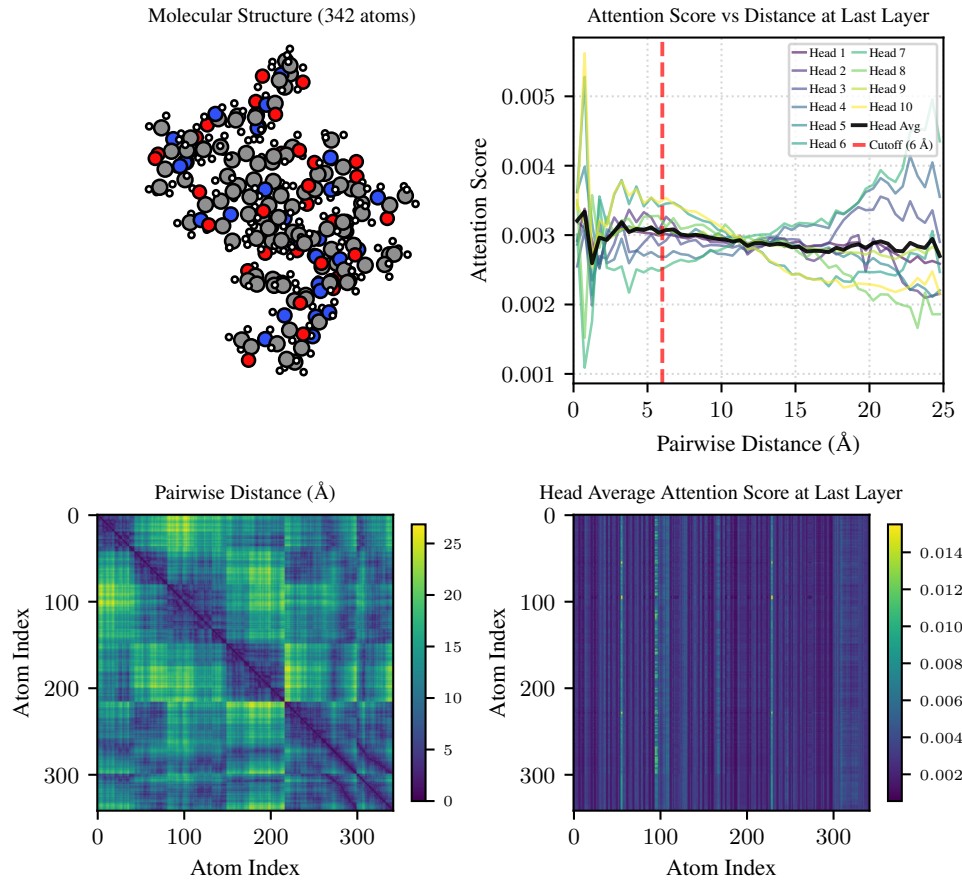

Figure 13: **Interpreting the All-to-all Node Attention.** Example protein pocket with 342 atoms in OMol Validation Set (top-left). Top-right: per-head attention weight in the last node-attention layer as a function of pairwise distance; thin lines = individual heads, thick black = head average. The red dashed line marks the local neighbor cutoff (6 Å). Bottom-left: pairwise distance matrix. Bottom-right: head-averaged attention matrix. Several heads allocate noticeable mass to pairs well beyond the local cutoff, and the attention heatmap shows column "hubs" that attract long-range attention across many rows.

# C  OC20 AND OMAT24 RESULTS

**Settings.**  We train AllScAIP on OC20 (240M) and OMat24 (100M) datasets. On OC20, we first train direct force for 3 epochs, and then conservative fine-tune for 1 epoch. We enable fp16 autocast during direct force training. On OMat24, we do 6 epochs direct force + 3 epochs conservative fine-tune, and it is full fp32 training.

**Results.**  We evaluate on OC20 using the Total Energy MAE (ID and OOD-Both) on the validation set, and OMat24 validation set (Table 5). We comparing with UMA (Wood et al., 2025), which is trained on more data (459M) and has a larger compute budget. On OC20, the medium model (AllScAIP-md-d.) shows competitive accuracy. It surpasses UMA-S in both Val/ID and Val/OOD-Both. The OOD/ID energy ratio ($\approx$1.6) is on par with UMA-M, indicating healthy generalization. On OMat24, the medium conservative model shows better performance than UMA-S. We did not devote substantial effort to model tuning for these two datasets.

Table 5: OC20 validation (Total Energy) and OMat24 validation results.

| | OC20 (Total Energy) | | | | | | OMat24 | | | |
| | Val ID | | | Val OOD-Both | | | Val | | | |
| Model | Energy | Force | F. Cos. | Energy | Force | F. Cos. | E./Atom | Forces | Stress | F. Cos. |
|---|---|---|---|---|---|---|---|---|---|---|
| UMA-S | 63.6 | 24.1 | 0.63 | 107.0 | 29.2 | 0.65 | 11.3 | 57.1 | 2.9 | 0.98 |
| UMA-M | 43.1 | 15.8 | 0.73 | 70.0 | 19.2 | 0.75 | 10.0 | 47.3 | 2.7 | 0.99 |
| UMA-L | 32.6 | 12.0 | 0.77 | 49.8 | 14.5 | 0.79 | 9.7 | 43.5 | 2.5 | 0.99 |
| eqV2-S | | | | | | | 11 | 49.2 | 2.4 | 0.985 |
| eqV2-M | | | | | | | 10 | 44.8 | 2.3 | 0.986 |
| eqV2-L | | | | | | | 9.6 | 43.1 | 2.3 | 0.987 |
| AllScAIP-sm-d. | 61.1 | 18.3 | 0.67 | 92.8 | 22.9 | 0.68 | 12.7 | 60.1 | - | 0.98 |
| AllScAIP-sm-ft-cons. | 72.3 | 22.5 | 0.64 | 120.1 | 27.3 | 0.64 | 12.4 | 56.0 | 2.8 | 0.98 |
| AllScAIP-md-d. | 59.3 | 17.6 | 0.69 | 92.2 | 21.8 | 0.70 | 11.4 | 56.6 | - | 0.98 |
| AllScAIP-md-ft-cons. | | | | | | | 10.7 | 54.3 | 2.7 | 0.98 |

# D  SPICE RESULTS

**Settings.**  To compare with other long-range MLIPs (e.g. LES (Kim et al., 2025)), we trained a small (34M), energy-conserving variant (`AllScAIP-sm-cons.`) on the SPICE dataset using the MACE-OFF splits (Eastman et al., 2023), following Kim et al. (2025) (we note that the SPICE dataset is included and recalculated at a higher level of theory in OMol25 (Levine et al., 2025)). We train for 300 epochs on H200 142GB with full fp32 precision.

**Results.**  We evaluate on seven held-out test subsets PubChem, DES370K monomers/dimers, Dipeptides, Solvated Amino Acids, Water, and QMugs. We report Energy/Atom MAE (meV; lower is better) and Force MAE (meV/Å; lower is better). Table 6 compares against MACE/MACELES (Kim et al., 2025), EScAIP (Qu & Krishnapriyan, 2024), and eSEN (Fu et al., 2025). We found that `AllScAIP-sm-cons` achieves the lowest overall energy and force errors across splits. These results indicate that the model recipe is useful in different dataset settings, and it shows superior performance over other long-range MLIP designs, such as MACELES (Kim et al., 2025).

Table 6: **SPICE test results.** Energy / Atom MAE (meV, ↓) and Force MAE (meV/Å, ↓) across different splits of SPICE. AllScAIP achieve the lowest overall energy and force errors.

| | MACE-L | | MACE-M | | MACELES-M | | EScAIP | | eSEN | | AllScAIP | |
| Dataset | E | F | E | F | E | F | E | F | E | F | E | F |
|---|---|---|---|---|---|---|---|---|---|---|---|---|
| PubChem | 0.88 | 14.75 | 0.91 | 20.57 | 0.71 | 17.24 | 0.53 | 5.86 | 0.15 | 4.21 | **0.13** | **3.21** |
| DES370K M. | 0.59 | 6.58 | 0.63 | 9.36 | 0.54 | 7.65 | 0.41 | 3.48 | 0.13 | 1.24 | **0.07** | **0.97** |
| DES370K D. | 0.54 | 6.62 | 0.58 | 9.02 | 0.47 | 7.48 | 0.38 | 2.18 | 0.15 | 2.12 | **0.06** | **0.95** |
| Dipeptides | 0.42 | 10.19 | 0.52 | 14.27 | 0.52 | 11.46 | 0.31 | 5.12 | 0.25 | 3.68 | **0.06** | **1.54** |
| Solvated A.A. | 0.98 | 19.43 | 1.21 | 23.26 | 0.82 | 18.37 | 0.61 | 11.52 | 0.25 | **3.68** | **0.13** | 4.86 |
| Water | 0.83 | 13.57 | 0.76 | 15.27 | 0.69 | 12.54 | 0.72 | 10.31 | 0.15 | 2.50 | **0.08** | **2.12** |
| QMugs | 0.45 | 16.93 | 0.69 | 23.58 | 0.76 | 20.11 | 0.41 | 8.74 | 0.12 | 3.78 | **0.11** | **2.73** |

# E   ADDITIONAL MD SIMULATIONS

## E.1   RESULTS ON MD22

To assess MD structural fidelity under distribution shift, we ran *zero-shot* NVT simulations on seven MD22 molecules (Chmiela et al., 2023) using the OMol-trained model (Figure 14) We computed the radial distribution $h(r)$ from the production segments and compared against reference trajectories generated with PBE+MBD, which a lower level of theory than OMol, hence not expected to coincide with our training target. Across systems, AllScAIP reproduces the same peak positions and overall shapes as eSEN-sm and UMA-md, indicating comparable equilibrium structure and no gross biases in intermolecular distances.

# F   DETAILED MODEL CONFIGURATIONS

Table 7: AllScAIP Model Configurations

| Model Size | sm_NeNo | md_NeNo | lg_NeNo | xlg_NeNo |
|---|---|---|---|---|
| hidden_size | 512 | 640 | 1024 | 2048 |
| num_layers | 6 | 10 | 8 | 12 |
| atten_num_heads | 8 | 10 | 16 | 32 |
| atten_name | | memory_efficient | | |
| activation | | gelu | | |
| normalization | | rmsnorm | | |
| node_direction_expansion_size | | 10 | | |
| edge_direction_expansion_size | | 6 | | |
| edge_distance_expansion_size | | 512 | | |
| attn_num_freq | | 32 | | |
| ffn_hidden_layer_multiplier | | 2 | | |
| output_hidden_layer_multiplier | | 2 | | |
| max_num_elements | | 110 | | |
| max_batch_size | | 96 | | |
| knn_soft | | True | | |
| knn_sigmoid_scale | | 0.2 | | |
| knn_lse_scale | | 0.1 | | |
| knn_use_low_mem | | True | | |
| distance_function | | gaussian | | |
| use_envelope | | True | | |
| mlp_dropout | | 0.0 | | |
| atten_dropout | | 0.0 | | |

## LLM USAGE STATEMENT

LLM is used for polishing the manuscript and organizing tables. Specifically, we use an LLM to refine language, improve readability, and enhance clarity.

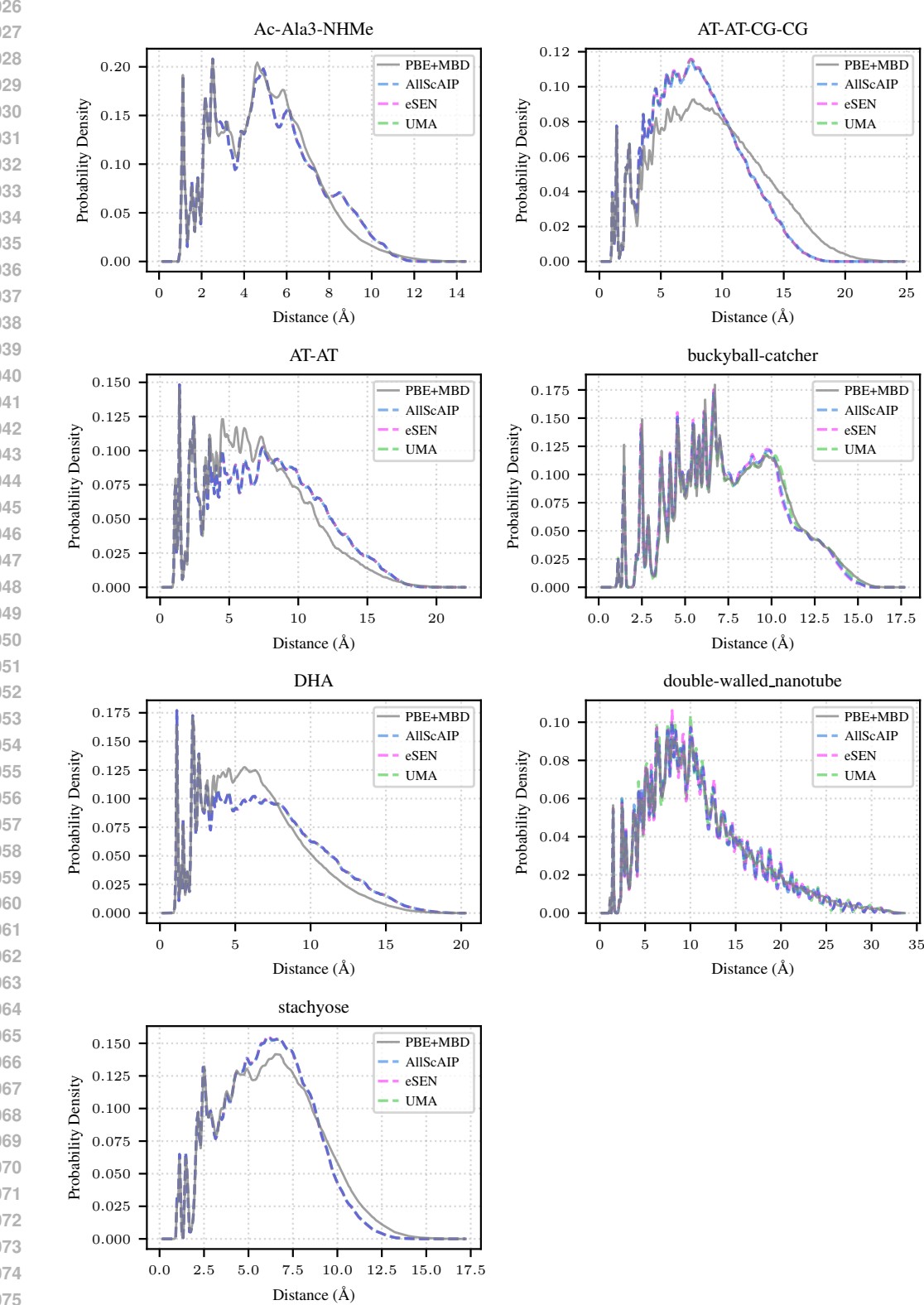

Figure 14: **Zero-shot MD on MD22: radial distribution** $h(r)$. NVT trajectories with OMol-trained AllScAIP on seven MD22 systems. Curves show $h(r)$ from the simulations compared with PBE+MBD reference runs. Although the reference level differs from OMol, AllScAIP tracks eSEN/UMA in peak locations and overall shape, indicating matched structural statistics.

