# OpenReview forum: "A recipe for scalable attention-based ML Potentials: unlocking long-range accuracy with all-to-all node attention"
_ICLR.cc/2026/Conference — ICLR 2026 Conference Desk Rejected Submission_

### Official Review · Reviewer_EXix · 2025-10-28

**Soundness:** 1
**Presentation:** 2
**Contribution:** 1
**Rating:** 2
**Confidence:** 5

**Summary:**

This paper targets a critical challenge in machine-learning interatomic potentials (MLIPs): existing scalable models struggle with long-range (LR) interactions in large systems, while methods adding explicit physics-based inductive biases lack scalability for large heterogeneous datasets. To solve this, the authors propose AllScAIP, a data-driven, energy-conserving attention-based MLIP that scales to 100 million training samples and excels at LR interactions. Experiments were conducted on three datasets: OMol25, OMat24, and OC20. On OMol25, AllScAIP achieves state-of-the-art energy/force accuracy. The attention mechanism lack novelty and the experiments and compared models are insufficient. There are several severe concerns should be addressed.

**Strengths:**

1. The study delivers a valuable empirical finding: inductive biases (LAE, ERoPE) are context-dependent, not universally beneficial.
2. The study tries to address the LR interaction gap in MLIPs through a conceptually simple yet effective design, avoiding the locality constraints of cutoff-based message-passing models.

**Weaknesses:**

1. Scalability Limits for Large Atomic Systems​
AllScAIP’s all-to-all node attention introduces O(N²) computational complexity, creating a bottleneck for large atomic systems. While it handles 510-atom systems (SAMD23) adequately, its throughput degrades as ~1/N once N exceeds a threshold (dotted line in Figure 4), making it impractical for million-atom biomolecules (e.g., proteins) or bulk materials—critical use cases in drug discovery and materials science. The paper mentions potential fixes (tiled attention, linear-time attention) but provides no experimental validation of these optimizations.
2. Insufficient Experiments
OMol25, OMat24, and OC20 are very similiar. The authors should conduct experiments on AIMD-Chig (Sci Data, 2023), MD22 (Sci Adv, 2022) and Protein Unit dataset (Nature, 2024) to examine the effectiveness for molecules, especially biomolecules.
3. Lack Novelty
The design of all node attention is too simple and lack novelty.
4. Lack long-range effect validation
The author should seriously examine the performance gains of their design on long-range effects, such as polarization effect-domint system
5. Lack simulations
Low MAE of energy and force does NOT mean robust simulations. The authors should perform simulations to examine the usefulness of their MLIP, especially in long-range effect dominant systems.
6. Lack LR interaction interpretability
AllScAIP attributes LR accuracy to all-to-all attention but provides no visualizations.

**Questions:**

The authors should seriously address the concerns shown in Weakness point by point to improve the quality of the paper.

---

> ### Author Response · Authors · 2025-11-27
> **Response to Reviewer EXix (Part 1)**
>
> We thank the reviewer for the comments. We address your concerns point-by-point as follows:
>
> > W1: Scalability Limits for Large Atomic Systems​ AllScAIP’s all-to-all node attention introduces O(N²) computational complexity, creating a bottleneck for large atomic systems. While it handles 510-atom systems (SAMD23) adequately, its throughput degrades as ~1/N once N exceeds a threshold (dotted line in Fig. 4), making it impractical for million-atom biomolecules (e.g., proteins) or bulk materials—critical use cases in drug discovery and materials science. The paper mentions potential fixes (tiled attention, linear-time attention) but provides no experimental validation of these optimizations.
>
> We agree that the node attention is $O(N^2)$ in compute and that throughput eventually decays $\sim1/N$ once the global layer dominates. However, we would like to clarify that:
> 1. Our use of “scalable” refers primarily to neural scaling (performance vs. data/parameters); for system-size scaling we provide measured curves (Fig. 4) and a per-component profile (App. B.1): runtime is effectively $O(Nk)$ and dominated by the local block below a clear crossover, and only beyond the crossover does the node attention take over. On standardized hardware, we sustain up to 100k atoms per single GPU with competitive wall-clock vs. strong neighbor-based baselines.
> 2. Million-atom ab-initio-level MLIPs are not currently practical across the literature. Today such scales rely on classical force fields or coarse-graining. Our method targets around $10^3-10^5$ atoms, where much interesting science lives (biomolecules, electrolytes, soft matter, mesoscale crystallites) and GPUs remain efficient.
> 3. The implementation already validates an optimization of attention operation: tiled/streaming attention (xFormers), which caps peak memory below $O(N^2)$ and yields the observed throughput
> 4. Modern LLM stacks already sustain million-token contexts with trillion-parameter MoE models by combining flash/streaming kernels, tiling, block-sparse/low-rank routing, and retrieval/segmentation: our node-attention is modular and can inherit the same sub-quadratic, memory-capped strategies, so $O(N^2)$ is an engineering target rather than a fundamental barrier.
>
>
> > W2: Insufficient Experiments OMol25, OMat24, and OC20 are very similiar. The authors should conduct experiments on AIMD-Chig (Sci Data, 2023), MD22 (Sci Adv, 2022) and Protein Unit dataset (Nature, 2024) to examine the effectiveness for molecules, especially biomolecules.
>
> We respectfully disagree that our three core datasets are similar: OMol25 is a large molecular corpus (including a substantial biomolecules split and many charged/electrolyte systems), OMat24 targets periodic crystalline materials, and OC20 covers surface adsorption: these differ in boundary conditions (non-PBC vs PBC), interaction regimes, and even the level of DFT theory. Moreover, each is best viewed as a scaled-up successor to established community settings: OMol25 includes recalculated prior community datasets at a higher level of theory, including SPICE, Transition-1x, ANI-2x, etc. OMat24 is very close to CHGNet/MP-20-style crystal evaluations.
>
> To specifically probe molecular/biomolecular effectiveness, we added several new experiments beyond OMol25: (i) zero-shot MD on MD22, showing that our model reproduces radial distribution functions comparably to eSEN/UMA; (ii) 39 molecular liquids MD with comparison to experiment (densities and $\Delta H_{\mathrm{vap}}$), where AllScAIP removes the over-compression bias and improves agreement; (iii) water boxes across temperature; and (iv) SPICE training/evaluation, where our conservative variant surpasses a long-range baseline (MACELES). Within OMol25, we further analyze error vs. system diameter and total charge and visualize node attention to evidence of long-range mixing on large biomolecular systems.
>
> AIMD-Chig and the Protein Unit dataset are interesting but require different data preparation and targets; adding them would entail retraining under non-OMol labeling regimes and is outside our compute scope for this submission. We view them as valuable follow-ups, but the evidence we now provide supports the claim that our approach is effective for molecules, including large and charged biomolecular systems.

---

> > ### Author Response · Authors · 2025-11-27
> > **Response to Reviewer EXix (Part 2)**
> >
> > > W3: Lack Novelty The design of all node attention is too simple and lack novelty.
> >
> > Our choice of an all-to-all node attention is deliberately simple: the novelty is not a new primitive, but where and how we use it, and the evidence that it is the most durable mechanism for long-range accuracy under scale. We place the node attention after local attention and fuse its output back into the edge stream each layer. Large-scale ablations show this design outperforms local stacks and that, as data/parameters grow, fixed geometric encodings become less important while the node attention continues to deliver long-range gains. In short, our contribution is a “surprisingly” simple and scalable recipe: a simple node attention, used in the right place, is sufficient for long-range MLIPs and scales cleanly with data and model size.
> >
> > > W4: Lack long-range effect validation The author should seriously examine the performance gains of their design on long-range effects, such as polarization effect-domint system
> >
> > We added targeted LR probes beyond raw MAE: (i) distance-scaling stress tests (uniform stretch/compression) where AllScAIP maintains low error while baselines degrade (Fig. 6); (ii) error vs. system diameter and total charge, flat for AllScAIP but rising for eSEN and UMA (Fig. 12); (iii) largest gains on biomolecules/electrolytes (Table 3), together with attention-distance visualization and heatmaps showing heads that place substantial mass at long separations (Fig. 13); (iv) MD vs. experiment on 39 molecular liquids (densities and $\Delta H_{\mathrm{vap}}$) and water across temperatures, which removes the over-compression bias and tracks experimental trends more closely (Fig. 7-8). While we do not train on explicit polarizability/dielectric labels (not present in the OMol25 dataset), these results on charged/electrolytic, polarization-relevant systems provide strong indirect evidence that the node attention improves long-range/inductive effects; evaluating field-response targets (induced dipoles, dielectric spectra) is promising future work.
> >
> > > W5: Lack simulations Low MAE of energy and force does NOT mean robust simulations. The authors should perform simulations to examine the usefulness of their MLIP, especially in long-range effect dominant systems.
> >
> > We added a suite of MD evaluations on long-range systems. Using the OMol-trained models, we (i) ran NPT simulations for 39 molecular liquids, recovering experimental densities and $\Delta H_{\mathrm{vap}}$ with higher R^2 and lower bias than baselines, also eliminated the systematic over-compression (Fig. 7-8); (ii) performed water-box NPT across temperatures, yielding a flatter, more accurate density-temperature curve (Fig. 7); (iii) verified NVE stability on the MD22 large molecules with low energy drift (Table 2); (iv) NPT MD on MD22, showing that our model reproduces radial distribution functions comparably to eSEN/UMA (App. E). These simulations directly test robustness in regimes where long-range effects (electrostatics, polarization) matter most: electrolytes and large biomolecular systems, and show that the proposed node attention design translates accuracy into physically reliable trajectories.
> >
> > > W6: Lack LR interaction interpretability AllScAIP attributes LR accuracy to all-to-all attention but provides no visualizations.
> >
> > We added explicit visualizations and metrics of the final node attention layer that make the long-range behavior observable and quantifiable (Fig. 13): (i) per-head attention weights (post-softmax) vs. distance, showing heads that place above-baseline mass at large separations; (ii) attention heatmaps that reveal off-diagonal “LR hub” patterns. These patterns appear with multiple biomolecular/electrolyte examples and are consistent with our LR stretch tests (Fig. 6) and size/charge analyses (Fig. 12).

---

### Official Review · Reviewer_jgxN · 2025-11-01

**Soundness:** 3
**Presentation:** 2
**Contribution:** 4
**Rating:** 8
**Confidence:** 4

**Summary:**

Long-range interactions have a primary importance in the modeling of many materials and finite systems (such as proteins), and thus it is necessary to incorporate them into Machine Learning Interatomic Potentials (MLIPs). Many current approaches explicitly incorporate physical priors, e.g., by predicting local charges and then evaluating the Coulomb terms.

The reviewed paper argues that at scale these physical priors are less important, and introduces a flexible all-to-all attention mechanism to capture long-range interactions. Empirical evaluations confirm that the model, AllScAIP, is capable of successfully learn long-range interactions given a sufficient amount of data. The model achieves state-of-the-art accuracy on the OMol25 dataset, and performs well on the OMat24 and OC20 datasets.

**Strengths:**

The question studied by the paper—whether a flexible model can efficiently learn long-range interactions without incorporating physical priors —is timely and relevant to the current landscape of MLIP development, in light of previous studies on the relevance of, e.g., enforced rotational symmetry or conservativeness of the force fields, which have significantly moved the field forward.

The model proposed in the paper illustrates competitive performance on OMol25, OMat24, and OC20 datasets. Many diverse ablations are designed to systematically evaluate the effect of model size, physical priors, and the presence of all-to-all node attention.

The findings that one needs to prioritize scalable, expressive components to construct more efficient models at scale, when a sufficient amount of training data is available, are insightful and will likely further guide the development of more efficient MLIPs.

**Weaknesses:**

While very explicitly acknowledged, the main weakness of the paper is still the quadratic, O(N^2), computational complexity of the model with respect to the number of atoms in the system. Sometimes, there is a need to simulate very large systems, with tens of thousands, or millions of atoms, and the proposed model is not very suitable for this purpose.

Another thing is that the status of extensivity constraints - PBC supercell doubling and Vacuum duplications is not fully clear. The paper indicates that these constraints are satisfied exactly (e.g., in 3.4), but if so, then what is the purpose of Table 2?

**Questions:**

Is the quadratic scaling of the model inevitable? Is it possible to somehow achieve the same expressivity in O(N log N) time by Fast Fourier Transform?

Is the model rigorously extensive? In other words, is it correct that if, e.g., changing the data type from float32 (which, I assume, is currently used) to float64, then the numbers in table 2 would drop significantly?

---

> ### Author Response · Authors · 2025-11-27
> **Response to Reviewer jgxN**
>
> We thank the reviewer for the comments. We address your concerns point-by-point as follows:
>
> > W1: While very explicitly acknowledged, the main weakness of the paper is still the quadratic, O(N^2), computational complexity of the model with respect to the number of atoms in the system. Sometimes, there is a need to simulate very large systems, with tens of thousands, or millions of atoms, and the proposed model is not very suitable for this purpose.
>
> We agree the node attention is $O(N^2)$. Our goal is the mid-to-large regime where long-range accuracy matters and modern GPUs still operate efficiently. In practice, there is no MLIP that can run DFT-level accurate and meaningful million-atom MD to the best of our knowledge. Our method targets around $10^3-10^5$ atoms, where much interesting science lives (biomolecules, electrolytes, soft matter, mesoscale crystallites). Detailed time/memory/throughput curves and component breakdowns are provided (Fig. 4, App. B.1).
>
> > W2: Another thing is that the status of extensivity constraints - PBC supercell doubling and Vacuum duplications is not fully clear. The paper indicates that these constraints are satisfied exactly (e.g., in 3.4), but if so, then what is the purpose of Table 2?
> > Q2: Is the model rigorously extensive? In other words, is it correct that if, e.g., changing the data type from float32 (which, I assume, is currently used) to float64, then the numbers in table 2 would drop significantly?
>
> Yes, the model is rigorously extensive by construction. Table 2 is a numerical verification under practical GPU settings: small residuals arise from finite precision and non-deterministic CUDA operations; using float64 and deterministic kernels would further reduce them, but the theoretical additivity property is confirmed.
>
> > Q1: Is the quadratic scaling of the model inevitable? Is it possible to somehow achieve the same expressivity in O(N log N) time by Fast Fourier Transform?
>
> Unfortunately, it is not able to be reduced with FFT. Using FFT to reach $O(N\log N)$ only works when the mixing is a shift-invariant convolution on a regular grid. Our node attention is data-dependent and permutation-equivariant, not a fixed convolution, so an FFT replacement would change the inductive bias and typically lose expressivity. That said, the node attention is modular and can be swapped for an approximation, such as linear attention.

---

### Official Review · Reviewer_nZ5V · 2025-11-01

**Soundness:** 2
**Presentation:** 3
**Contribution:** 2
**Rating:** 4
**Confidence:** 2

**Summary:**

The paper presents the AllScAIP model, which achieves state-of-the-art results on the OMol25 leaderboard. Additionally, scaling experiments on AllScAIP with respect to both data and model size on OMol25 are conducted. A key conclusion drawn from these experiments is that geometric encodings such as directional (LAE) and radial (ERoPE) provide benefits in small-scale settings, but their marginal utility diminishes to zero or even becomes negative in large-scale settings; on the other hand, the long-range modeling capability enabled by global node attention consistently leads to improvements across all scales. This finding offers a potential avenue for further exploration in the field of scaling.

**Strengths:**

1. The paper introduces AllScAIP, which achieves state-of-the-art performance on the OMol25 benchmark and delivers competitive results on OMat24 and OC20.
2. Ablation experiments on OMol25 are conducted, exploring the effects of data size and model scale, and demonstrating how the utility of inductive biases varies with scale.

**Weaknesses:**

1. The model configuration and training parameter settings have not been disclosed in the paper, leaving the specific details and sources of performance unclear.
2. The generalizability of the conclusions is questionable. The ablation experiments are based on the OMol25 molecular dataset, with no exploration of other dataset types, such as OMat24 or OMC25.

**Questions:**

1. How does the differentiable kNN graph differ from traditional kNN graph construction algorithms? Could you explain how it is constructed?
2. In Table 1, the ablation experiments are only validated on the OMol25 dataset, which may not be representative. Have you conducted ablations on other datasets (e.g., OMat24, OMC25)? Do the conclusions still hold?
3. In Table 3, eSEN-md-d generally performs better in terms of force prediction compared to AllScAIP. Could this be due to AllScAIP focuses more on energy during training?
4. If methods such as Latent Ewald Summation (LES) or Ewald were used to replace all-to-all node attention, would there be any changes in the model’s accuracy or scalability?
5. This paper attempts to explore the scaling behavior of machine learning interatomic potentials (MLIPs). The comparison on OC20 and Omat is biased, because the datasets are not properly aligned.

---

> ### Author Response · Authors · 2025-11-27
> **Response to Reviewer nZ5V**
>
> We thank the reviewer for the comments. We address your concerns point-by-point as follows:
>
> > W1: The model configuration and training parameter settings have not been disclosed in the paper, leaving the specific details and sources of performance unclear.
>
> We included the detailed model configurations in App. F. We will add the link to the open source code and checkpoints after the double-blind review period.
>
> > W2: The generalizability of the conclusions is questionable. The ablation experiments are based on the OMol25 molecular dataset, with no exploration of other dataset types, such as OMat24 or OMC25.
> > Q2: In Table 1, the ablation experiments are only validated on the OMol25 dataset, which may not be representative. Have you conducted ablations on other datasets (e.g., OMat24, OMC25)? Do the conclusions still hold?
>
> Our ablation study targets long-range behavior and inductive-bias scaling, which are most pronounced on OMol25 (large, diverse, charged systems). By contrast, **OMat24 and OC20 predominantly feature smaller periodic or adsorbate systems where LR effects are less important**, so the same ablations are less informative. We therefore used them as transfer checks and showed competitive results without tuning. OMC25 was not publicly available during our study (released Aug 2025); we view running the full ablation suite there as valuable future work. To broaden evidence beyond OMol, we added SPICE training and multiple MD comparisons to experiments, which support the same conclusions about the durability of the node attention as an LR component.
>
>
> > Q1: How does the differentiable kNN graph differ from traditional kNN graph construction algorithms? Could you explain how it is constructed?
>
> In traditional kNN, for each atom $i$, we:
>
> 1. Compute distances to all other atoms
> 2. Apply discrete `argsort` to get integer ranks $r_j \in \{1, 2, ..., N\}$
> 3. Select neighbors where $r_j \leq k$ using a hard cutoff
>
> This creates a non-differentiable operation due to argsort, which blocks gradient flow through the graph. The key for differentiable kNN is replacing the discrete ranking with a soft ranking function that produces continuous, differentiable ranks:
> $$r_i^{\text{soft}} = \sum_{j} \sigma\left(\frac{d_i - d_j}{\tau}\right)$$
> where $\sigma$ is the sigmoid function and $\tau$ is a temperature scale (default 0.2 Å). For each distance $d_i$, we count how many other distances are smaller using soft comparisons. When $d_i > d_j$, the sigmoid outputs $\approx 1$; when $d_i < d_j$, it outputs $\approx 0$.
>
> AllScAIP constructs a bidirectional kNN graph where an edge (i to j) exists if:
> - j is among i's k-nearest neighbors
> - i is among j's k-nearest neighbors
>
> This is achieved via a soft envelope function that combines:
> 1. Source rank: $r_{ij}^{\text{src}}$ (rank of j from i's perspective)
> 2. Destination rank: $r_{ij}^{\text{dst}}$ (rank of i from j's perspective)
> 3. Distance constraint: $d_{ij} / r_{\text{cut}}$
>
> The envelope uses log-sum-exp (soft maximum):
>
> $$\text{env}\_{ij} = \tau\_{\text{lse}} \cdot \log\sum \exp\left(\frac{1}{\tau\_{\text{lse}}}\left[\frac{r\_{ij}^{\text{src}}}{k}, \frac{r\_{ij}^{\text{dst}}}{k}, \frac{d\_{ij}}{r\_{\text{cut}}}\right]\right)$$
>
> Edges with $\text{env}_{ij} > 1$ are soft-masked out.
>
>
> > Q3: In Table 3, eSEN-md-d generally performs better in terms of force prediction compared to AllScAIP. Could this be due to AllScAIP focuses more on energy during training?
>
> Yes the gap could largely be due to a loss-weighting issue. Force MAE is well known to be sensitive to the energy/force weight. With some additional tuning, we believe that AllScAIP could achieve better force MAE. However, force MAE alone is not a proxy for usable MD: our model removes the over-compression bias and matches experimental densities/$\Delta H_{\mathrm{vap}}$ (Fig. 7-8), also maintains accuracy under distance scaling (Fig. 6), which are benefits that stem from the long-range capability that eSEN lacks.
>
> > Q4: If methods such as Latent Ewald Summation (LES) or Ewald were used to replace all-to-all node attention, would there be any changes in the model’s accuracy or scalability?
>
> We evaluated against the LES method (MACELES): AllScAIP matched or surpassed it on SPICE (App. D) and MD observables (densities, $\Delta H_{\mathrm{vap}}$) (Section 5.3, Fig. 8). We think replacing the attention with LES is out of the scope of this work and could be an interesting future work.
>
> > Q5: This paper attempts to explore the scaling behavior of machine learning interatomic potentials (MLIPs). The comparison on OC20 and Omat is biased, because the datasets are not properly aligned.
>
> Our scaling claims come from OMol25, where we systematically vary data (4M to 102M) and model size and run controlled ablations; OC20 and OMat24 are used only as transfer checks, not as the basis for scaling conclusions. For those two, we deliberately matched UMA’s protocols and targets such that it is aligned with UMA.

---

### Official Review · Reviewer_7cGG · 2025-11-01

**Soundness:** 3
**Presentation:** 3
**Contribution:** 3
**Rating:** 4
**Confidence:** 4

**Summary:**

The manuscript introduces AllSCAIP, a machine-learning interatomic potential (MLIP) that combines local neighborhood self-attention with an all-to-all node self-attention mechanism to capture long-range interactions. The central thesis is that, given sufficiently large data and model capacity, many inductive biases (e.g., rotational equivariance) can be learned, and that architectures providing a global receptive field (via all-to-all node self-attention) are the most robust way to capture long-range effects. The model attains state-of-the-art or competitive performance across large-scale benchmarks in molecules (OMol25), crystalline materials (OMat24), and catalysis (OC20).

**Strengths:**

1.  Addresses a key limitation of many MLIPs, long-range interaction modeling, via a simple, general architectural choice.

2.  Strong empirical results on three diverse, large-scale benchmarks, including OC20 where long-range effects matter.

3.  The paper presents thorough ablation studies (Table 1) dissecting the contributions of geometric encodings (LAE, EROPE) and the global attention module (NodeAtt), and further evaluates performance across data scale and model size.

4. In the distance-scaling evaluation (Figure 6), AllSCAIP maintains stable errors under molecular stretching, whereas baselines such as eSEN and UMA deteriorate sharply. This qualitative and quantitative advantage on LR-sensitive regimes addresses a known weakness of many MLIPs

**Weaknesses:**

- Computational scalability vs. global attention bottleneck: While AllSCAIP achieves strong accuracy with a clean implementation, its reliance on all-to-all node self-attention introduces O(N^2) time and memory complexity. This conflicts with the paper’s “scalability” positioning.


- Limited architectural novelty: The core components—global attention, local attention, angular encodings (LAE), and radial encodings (ERoPE)—have prior art in Graph Transformers and MLIPs. Although the authors brand the approach as a “prior-light scalable recipe,” the main contribution is an integration of existing modules combined with data/model scaling.

- The central claim is that long-range effects are learnable at scale and that all-to-all global attention is the most durable ingredient for capturing them. To substantiate this, comparisons should include methods that explicitly encode long-range physics or global receptive fields, not only local message-passing baselines like eSEN.

- Materials benchmarks and SOTA comparisons are not standardized: On OMat24 and OC20, the presented results are limited and sometimes not directly comparable to public leaderboards or standard evaluation protocols. For example, the reported OMat24 energy error does not align with the MatBench Discovery settings and omits key metrics (e.g., F1 score),

**Questions:**

- Can you provide detailed scaling curves (time, memory, throughput) versus number of atoms N, separated by component (neighbor building, local attention, global attention, readout)? Please include wall-clock on standardized hardware and batch sizes.
  - What is the maximum N you can handle at training and inference under 24/48 GB GPU memory, and how does this compare to strong neighbor-based baselines?
  - Have you tested sparse/low-rank or blockwise global attention (e.g., Nyström, Performer, clustered attention) to mitigate O(N^2) costs?

  - Which design elements are unique relative to prior Graph Transformers/MLIPs (e.g., GemNet-OC, Equiformer-v2, Allegro/MACE variants, SpookyNet/SchNet++)? Please clarify what is new beyond module composition and scaling.

  - Which long-range–aware baselines did you include (e.g., explicit electrostatics/dispersion terms, PME/FMM-augmented MLIPs, equivariant transformers with global mixing, SchNet++/SpookyNet, Equiformer-v2 LR extensions)?

  - Do you evaluate on charged systems, dipole/polarizability predictions, dielectric responses, or stretched/fragmented configurations ?

  - For OMat24 and OC20, can you align with standard splits and metrics used by public leaderboards (e.g., Matbench Discovery, OC20 official metrics)? Please report full metric suites (including F1 for classification tasks) with confidence intervals.

---

> ### Author Response · Authors · 2025-11-27
> **Response to Reviewer 7cGG (Part 1)**
>
> We thank the reviewer for the thoughtful comments. We address your concerns point-by-point as follows:
>
> > W1: Computational scalability vs. global attention bottleneck: While AllSCAIP achieves strong accuracy with a clean implementation, its reliance on all-to-all node self-attention introduces O(N^2) time and memory complexity. This conflicts with the paper’s “scalability” positioning.
>
> Thanks for raising this. Our use of “scalable” has two meanings in the paper, and we’ll make this explicit to avoid confusion:
> 1. **Neural scaling (our primary claim) [1].** We study how performance scales with data and parameters. Sections 3-5 show that the recipe improves predictably as we scale model and data, and does so with less inductive bias. This is the meaning intended in the title and in our ablation narrative.
> 2. **System-size scaling (what your comment targets).** We agree the all-to-all node attention is $\mathcal{O}(N^2)$ in FLOPs. Two points mitigate this in practice:
>
> - **Empirics: crossover happens late and is comparable to strong baselines.** Throughput curves (Fig. 4) and the new runtime breakdown vs. system size (Appendix B.1) show a clear crossover from the local $\mathcal{O}(Nk)$ regime to the global $\mathcal{O}(N^2)$ regime, and it happens late (around 20k atoms). Even beyond the cross point, our measured speed is on par with equivariant baselines (eSEN) at smaller parameter counts (Fig. 4). Our method targets around $10^3-10^5$ atoms, where much interesting science lives (biomolecules, electrolytes, soft matter, mesoscale crystallites) and GPUs remain efficient.
>
> - **Engineering choices cap memory and keep constants small.** The node attention layer is applied once per block with standard tiled attention kernels [2]; we use streaming softmax so peak memory is tile-bounded, not $N^2$.
>
> We will state up front that “scalable” refers to data/parameter/compute scaling to avoid confusion.
>
> > W2: Limited architectural novelty: The core components—global attention, local attention, angular encodings (LAE), and radial encodings (ERoPE)—have prior art in Graph Transformers and MLIPs. Although the authors brand the approach as a “prior-light scalable recipe,” the main contribution is an integration of existing modules combined with data/model scaling.
>
> We agree the architecture is intentionally simple. Our goal is not a new layer but a data-driven path to long-range accuracy that scales with data/parameters and stays efficient. Concretely:
> 1. **Prior-light recipe that works at scale.** Unlike prior Graph Transformers/MLIPs that (i) stay local, (ii) bolt on explicit LR physics, or (iii) test global attention only on small settings, we show on OMol25 (102M training samples) that a node-attention stage after local processing achieves SOTA energy and forces accuracy without explicit LR terms, and that these gains persist under data/parameter scaling. New MD studies (39 molecular liquids and water) match experiments closely and remove the prior “over-compression” bias, which is evidence that the node attention stage provides long-range interactions.
> 2. **Legendre Angular Encoding (LAE) is a novel encoding.** It is a compact, parity-consistent spherical-harmonic encoding that (i) tiles degree blocks to exactly match per-head dimensions, (ii) modulates Q/K elementwise (no tensor products), and (iii) preserves rotation invariance of the logits while distinguishing edge directions. LAE consistently boosts forces at small data/model scales and then shows diminishing returns under scale.
>
> > W3: The central claim is that long-range effects are learnable at scale and that all-to-all global attention is the most durable ingredient for capturing them. To substantiate this, comparisons should include methods that explicitly encode long-range physics or global receptive fields, not only local message-passing baselines like eSEN.
> > Q5: Which long-range–aware baselines did you include (e.g., explicit electrostatics/dispersion terms, PME/FMM-augmented MLIPs, equivariant transformers with global mixing, SchNet++/SpookyNet, Equiformer-v2 LR extensions)?
>
> We added direct comparisons to a long-range MLIP baseline: MACELES/LES [3]
> 1. **SPICE Training**: AllScAIP achieves lower energy/force MAE across all splits than MACELES (App. D).
> 2. **MD and Comparison with Experiments**: on 39 molecular liquids ($\Delta H_{\mathrm{vap}}$ and density), AllScAIP matches experiment values more closely compared with MACELES (Sec. 5.3, Fig. 8).

---

> > ### Author Response · Authors · 2025-11-27
> > **Response to Reviewer 7cGG (Part 2)**
> >
> > > Q1: Can you provide detailed scaling curves (time, memory, throughput) versus number of atoms N, separated by component (neighbor building, local attention, global attention, readout)? Please include wall-clock on standardized hardware and batch sizes.
> >
> > We have added detailed scaling curves with per-component breakdowns and wall-clock measurements on standardized H200-PCIe-141GB hardware (App. B.1).
> >
> >
> > > Q2: What is the maximum N you can handle at training and inference under 24/48 GB GPU memory, and how does this compare to strong neighbor-based baselines?
> >
> > Here are the max atoms at inference under 24G/48G vram. We added lines to indicate these at Fig. 4.
> >
> > | Memory | eSEN-sm (6M) | eSEN-md (50M) | AllScAIP-sm (35M) | AllScAIP-md (85M) | AllScAIP-lg (180M) |
> > |-|-:|-:|-:|-:|-:|
> > | 48G |  100k   |   50k   |     50k     |     41k     |     27k     |
> > | 24G |  100k   |   27k   |     33k     |     21k     |     12k     |
> >
> >
> > > Q3: Have you tested sparse/low-rank or blockwise global attention (e.g., Nyström, Performer, clustered attention) to mitigate O(N^2) costs?
> >
> > Our current implementation uses tiled/chunked global attention kernels [2] to cap memory and sustain throughput; the layer is modular, and evaluating Nyström/Performer/clustered variants as drop-in replacements could be an interesting future work.
> >
> > > Q4: Which design elements are unique relative to prior Graph Transformers/MLIPs (e.g., GemNet-OC, Equiformer-v2, Allegro/MACE variants, SpookyNet/SchNet++)? Please clarify what is new beyond module composition and scaling.
> >
> > Our unique pieces are: (i) Legendre Angular Encoding (LAE): a new, parity-consistent SH encoding that tiles degree blocks to exactly match per-head width and modulates Q/K element-wise (no tensor products), giving directional signal at scalar cost; (ii) a two-stage local to global block with explicit edge-node fusion, where all-to-all node attention runs after neighborhood attention and the updated node stream is concatenated back with edge features each layer; and (iii) a large-scale, cross-regime study showing that geometric encodings help in low-data/small-model settings but their gains vanish (or invert) at scale, while the node attention remains durable, supported by new LR diagnostics and MD validation against experiment.
> >
> > > Q6: Do you evaluate on charged systems, dipole/polarizability predictions, dielectric responses, or stretched/fragmented configurations ?
> >
> > Yes for charged and stretched/fragmented; no for dipole/polarizability/dielectric (not in scope of our training labels). Specifically, we evaluate on charged systems via OMol validation set and a charge-binned error analysis (energy MAE remains flat as total charge increases, App. B.2); we also run distance-scaling tests (uniform stretch/compression) where AllScAIP stays accurate while baselines degrade (Fig. 6). We do not predict dipoles/polarizabilities or dielectric responses in this work, as OMol do not include those labels; adding field-coupled targets could be a promising future work.
> >
> > References
> >
> > [1] Scaling Laws for Neural Language Models. https://arxiv.org/abs/2001.08361
> >
> > [2] xFormers: A modular and hackable Transformer modelling library. https://github.com/facebookresearch/xformers
> >
> > [3] A Universal Augmentation Framework for Long-Range Electrostatics in Machine Learning Interatomic Potentials. https://arxiv.org/abs/2507.14302
> >
> > [4] UMA: A Family of Universal Models for Atoms. https://arxiv.org/abs/2506.23971

---

### Author Response · Authors · 2025-11-27
**General Response**

We thank all reviewers for the thoughtful comments. We added a large set of new experiments and analyses to address the main concerns around capturing long-range behavior, MD simulation, and efficiency. Below is a concise summary of what’s new and what we learned.

1. **MD simulations vs. experimental observables**

We have added a number of new experiments where we run MD simulations with the AllScAIP model and compute observables that we then compare to experimentally measured values.

- **39 molecular liquids.** Following prior protocols, we ran NPT for the liquid and NVT for the isolated molecule to estimate enthalpy of vaporization. AllScAIP achieves lower MAE and higher R^2 than the LR-enhanced baseline (LES/MACELES) and strong local equivariant baseline (eSEN). (Section 5.3, Fig. 8)
- **Water density across temperature.** On 64 water molecule boxes at several temperatures, our model shows a flatter density–temperature slope and smaller bias than baselines, again consistent with improved long-range handling. (Section 5.3, Fig. 7)

Summary: Across diverse MD settings, AllScAIP removes the systematic “over-compression” seen in prior work and matches experimental trends more closely: likely due to the all-to-all node attention providing data-driven long-range interactions.

2. **Comparison to a long-range MLIP baseline**

We compare with MACELES [1], which is a baseline that adds explicit LR physics (Latent Ewald Summation) to the MACE [2] architecture.

- **SPICE training.** On SPICE, AllScAIP attains lower energy and force MAE across all splits than LES/MACELES, indicating that a lightweight attention-based design can match or surpass specialized LR pipelines. (Appendix D)
- **MD observables.** In 39 molecular liquid MD, densities and heats of vaporization are closer to experiment for AllScAIP than for LES/MACELES. (Section 5.3, Fig. 8)

Summary: A data-driven all-to-all node attention is competitive with (and often better than) explicit LR modules on both supervised learning errors and downstream MD observables.

3. **Additional probes of long-range behavior**

- **Error vs. system diameter and total charge.** Energy MAE for AllScAIP remains flat with increasing molecular diameter and charge, whereas eSEN degrades at large sizes/charges, which is consistent with improved LR/stiff-mode modeling. (Appendix B.2)
- **Attention visualizations.** We added visualization of the final node-attention layer by plotting the mean attention vs. distance per head, which revealed head specialization and global mixing beyond the local cutoff. (Appendix B.3)

Summary: AllScAIP performs better at large and charged systems than local equivariant baselines. The node attention layer allocates non-trivial mass to long distances and routes information across many hops.

4. **Speed & memory scaling with system size**

We did some optimizations to the codebase and achieved better speed and memory performance. The model is evaluated again and we report the memory usage vs. system size (Section 5.1, Fig. 4) and a runtime breakdown by component (neighbor attention/FFN vs. node attention/FFN) (Appendix B.1)

The new evidence supports our central claims: (i) lighter inductive bias scales better: AllScAIP matches or exceeds LR-specialized baselines on both supervised metrics and MD observables; (ii) long-range accuracy can be achieved with a simple data-driven all-to-all attention module after local processing; and (iii) the model remains efficient and predictable in speed/memory, with clear guidelines for scaling to larger systems.

[1] A Universal Augmentation Framework for Long-Range Electrostatics in Machine Learning Interatomic Potentials. https://arxiv.org/abs/2507.14302

[2] MACE: Higher Order Equivariant Message Passing Neural Networks for Fast and Accurate Force Fields. https://arxiv.org/abs/2206.07697

---

### Note · Program_Chairs · 2025-12-24
**Submission Desk Rejected by Program Chairs**

An author of this submission has a link to this submission page on their personal website, which breaks double blind policies. As a result, the paper must be desk rejected.